# Manifold Interpolating Optimal-Transport Flows for Trajectory Inference

Guillaume Huguet[1][*]   D.S. Magruder[2][*]   Alexander Tong[1][*]   Oluwadamilola Fasina[2]

Manik Kuchroo[2]   Guy Wolf[1][†]   Smita Krishnaswamy[2][†]

[1]Université de Montréal; Mila - Quebec AI Institute    [2] Yale University

## Abstract

We present a method called Manifold Interpolating Optimal-Transport Flow (MIOFlow) that learns stochastic, continuous population dynamics from static snapshot samples taken at sporadic timepoints. MIOFlow combines dynamic models, manifold learning, and optimal transport by training neural ordinary differential equations (Neural ODE) to interpolate between static population snapshots as penalized by optimal transport with manifold ground distance. Further, we ensure that the flow follows the geometry by operating in the latent space of an autoencoder that we call a geodesic autoencoder (GAE). In GAE the latent space distance between points is regularized to match a novel multiscale geodesic distance on the data manifold that we define. We show that this method is superior to normalizing flows, Schrödinger bridges and other generative models that are designed to flow from noise to data in terms of interpolating between populations. Theoretically, we link these trajectories with dynamic optimal transport. We evaluate our method on simulated data with bifurcations and merges, as well as scRNA-seq data from embryoid body differentiation, and acute myeloid leukemia treatment.

## 1   Introduction

Here, we tackle the problem of continuous dynamics of probability distributions defined on a data manifold. Data from naturalistic systems are often modeled as generated from an underlying low dimensional manifold embedded in a high dimensional measurement space. Termed the manifold hypothesis, this assumption has led to many successful models of biological, chemical, and physical systems. Measurements in such systems are increasingly high dimensional. For instance, in single cell data the entire transcriptomic profile of the cell is measured with each mRNA or gene species a dimension. However, because of informational redundancy between these genes, the intrinsic dimensionality of the data is low dimensional. Mathematically, a Riemannian manifold is a good model for such a system, and much of manifold learning literature including, diffusion maps [9], Laplacian eigenmaps [2], and the wider graph signal processing literature, has focused on learning this structure via data graphs [31, 32] and autoencoders [30, 11].

Building on this literature, we consider snapshot measurements of cellular populations over time. Current technology cannot follow a cell in time as the measurements are destructive. We frame this as learning a population flow on a data manifold. Recently several neural networks that perform flows, or transports between probability distributions, have been proposed in the literature. However, most of these works have focused on generative modeling: i.e., flowing from a noise distribution such as a Gaussian distribution to a data distribution in order to generate data. Examples include score-based

---

[*]Equal contribution

[†]Corresponding authors: `guy.wolf@umontreal.ca` and `smita.krishnawamy@yale.edu`

generative matching [41, 42], diffusion models [21], Schrödinger bridges [33, 10], and continuous normalizing flows (CNF) [7, 17]. However, here we focus on learning continuous dynamics of such systems using static snapshot measurements, with a key task being interpolation in unmeasured times, as well as inference of individual trajectories that follow the manifold structure.

To continuously interpolate populations over time on data with manifold structure, we propose Manifold Interpolation Optimal-transport Flow (MIOFlow)[3], a new framework for flows based on dynamical optimal transport in a manifold embedding. MIOFlow uses a neural ordinary differential equations (Neural ODE) [7] to transport a sampling of high dimensional data points between time-points such that 1. the transport occurs on a manifold defined by samples, 2. the transport is penalized to agree with measured timepoints using Wasserstein, 3. the transport is inherently stochastic.

Works tackling a similar problem include TrajectoryNet [44], dynamics modeled by a recurrent neural network [20], and Waddington-OT [38]. TrajectoryNet is based on a continuous normalizing flow where the magnitude of the derivative is penalized to create a more optimal flow. However, this approach suffers from several drawbacks. The first drawback is the requirement of starting from a Gaussian distribution. It can be difficult to match distributions from the real world to Gaussian distributions in an interpolating sense, all intermediate distributions must be penalized to start from a Gaussian rather than being trained to flow from one to the next. Second, continuous normalizing flows are deterministic. Hence, to model the intrinsic stochasticity in biology, we have to force chaotic dynamics starting from slight noise added to any initial distributions. Third, continuous normalizing flow models in $k$ dimensions require calculating the trace of the Jacobian, which requires $O(k^2)$ operations to compute [7], making our method $k$ times faster per function evaluation.

Additionally, unlike normalizing flows which operate in ambient data dimensions, we focus on flows on a data manifold. We feature a two-pronged approach with which to enforce this. First, propose the *Geodesic Autoencoder* to embed the data such that distances in the latent space match a new manifold distance called diffusion geodesic distance. Second, we penalize the transport using a manifold optimal transport method [34, 50]. These two steps enforce flows on the manifold, whereas in earlier works, such as TrajectoryNet, the KL-divergence is used to match distributions, and in [27] access to the metric tensor of a Riemannian manifold is required.

The main contributions of our work include:

- The MIOFlow framework for efficiently learning continuous stochastic dynamics of static snapshot data based on Neural ODEs that implements optimal transport flows on a data manifold.
- The Geodesic Autoencoder to create a manifold embedding in the latent space of the autoencoder.
- A new multiscale manifold distance called diffusion geodesic distance, and theoretical results showing its convergence to a geodesic on the manifold in the limit of infinitely many points.
- Empirical results showing that our flows can model divergent trajectories on toy data and on single-cell developmental and treatment response data.

## 2   Preliminaries and Background

**Problem Formulation and Notation**   We consider a distribution $\mu_t$ over $\mathbb{R}^k$ evolving over time $t \in \mathbb{R}$, from which we only observe samples from a finite set of $T$ distributions $\{\mu_i\}_{i=0}^{T-1}$. For each time $t$, we observe a sample $\mathsf{X}_t \sim \mu_t$ of size $n_t$. We note the set of all observations $\mathsf{X}$ of size $n := \sum_i n_i$. We also want to characterize the evolution of the support of $\mu_t$. We aim to define a trajectory from an initial points $X_0 \sim \mu_0$ to $X_{T-1} \sim \mu_{T-1}$, given the intermediate conditions $X_2 \sim \mu_2, \ldots, X_{T-1} \sim \mu_{T-1}$. We are thus interested in matching distributions given a set of $T$ samples $\{\mathsf{X}_i\}_{i=1}^{T-1}$, and initial condition $\mathsf{X}_0$.

In the following, we assume that the distributions are absolutely continuous with respect to the Lebesgue measure, and we use the same notation for the distribution and its density. We note the equivalence between two distances $d_1 \simeq d_2$. We assume that all Stochastic Differential Equations (SDE) admit a solution [15]. All proofs are presented in the supplementary material.

**Optimal Transport**   In this section, we provide a brief overview of optimal transport [34, 50], our primary approach for interpolating between distributions using a neural ODE. We consider two distributions $\mu$ and $\nu$ defined on $\mathcal{X}$ and $\mathcal{Y}$, and $\Pi(\mu, \nu)$ the set of joint distributions on $\mathcal{X} \times \mathcal{Y}$ with

---

[3]Code is available here: `https://github.com/KrishnaswamyLab/MIOFlow`

marginals $\mu$ and $\nu$, i.e. $\pi(dx, \mathcal{Y}) = \mu(dx)$ and $\pi(\mathcal{X}, dy) = \nu(dy)$ for $\pi \in \Pi(\mu, \nu)$. The transport plan $\pi$ moves the mass from $\mu$ to $\nu$, where the cost of moving a unit mass from the initial $x \in \mathcal{X}$ to the final $y \in \mathcal{Y}$ is $d(x, y)$. This formulation gives rise to the $p$-Wasserstein distance

$$W_p(\mu, \nu)^p := \inf_{\pi \in \Pi(\mu, \nu)} \int_{\mathcal{X} \times \mathcal{Y}} d(x, y)^p \pi(dx, dy),$$

where $p \in [1, \infty)$. Suppose $\mathcal{X} = \mathcal{Y} = \mathbb{R}^k$, and $d(x, y) := \|x - y\|_2$ then Benamou and Brenier [3] provide a *dynamic formulation* of the optimal transport problem. For simplicity, we assume a fix time interval $[0, 1]$, but note that the following holds for any time interval $[t_0, t_1]$. The transport plan is replaced by a time-evolving distribution $\rho_t$, such that $\rho_0 = \mu$ and $\rho_1 = \nu$ and that satisfies the continuity equation $\partial_t \rho_t + \nabla \cdot (\rho_t v) = 0$, where $\nabla \cdot$ is the divergence operator. The movement of mass is described by a time-evolving vector field $v(x, t)$. When $\rho_t$ satisfies these conditions, Benamou and Brenier [3] show that

$$W_2(\mu, \nu)^2 = \inf_{(\rho_t, v)} \int_0^1 \int_{\mathbb{R}^k} \|v(x, t)\|_2^2 \rho_t(dx) dt. \tag{1}$$

This formulation arises from the field of fluid mechanics; the optimal vector field is the divergence of a pressureless potential flow. We can also view the problem on the path space of $\mathbb{R}^k$

$$W_2(\mu, \nu)^2 = \inf_{X_t} \mathbb{E}\left[ \int_0^1 \|f(X_t, t)\|_2^2 dt \right] \text{ s.t. } dX_t = f(X_t, t)dt, \ X_0 \sim \mu, \ X_1 \sim \nu, \tag{2}$$

where the infimum is over all absolutely continuous stochastic path $X_t$ (see [28]).

**Adding Diffusion**    For various applications, it is interesting to incorporate a diffusion term in the paths or trajectories. These types of flows are often used in control dynamics [1, 29, 16], and mean field games [24]. We consider an SDE $dX_t = f(X_t, t)dt + \sqrt{\sigma} dB_t$, where $B_t$ is a standard Brownian motion. To model this new dynamic, one can replace the continuity equation with the Fokker–Planck equation of this SDE with diffusion, i.e. $\partial_t \rho_t + \nabla \cdot (\rho_t v) = \Delta(\sigma \rho_t / 2)$. From this formulation, one can also retrieve the Benamou-Brenier optimal transport. Indeed, Mikami [28] established the convergence to the $W_2$ when the diffusion term goes to zero. Similar to (2), we can phrase the problem using an SDE

$$\inf_f \mathbb{E}\left[ \int_0^1 \|f(X_t, t)\|_2^2 dt \right] \text{ s.t. } dX_t = f(X_t, t)dt + \sqrt{\sigma} dB_t, \ X_0 \sim \mu, \ X_1 \sim \nu. \tag{3}$$

The two previous formulations admit the same entropic interpolation $\rho_t$ [33, 28]. In this paper, we utilize such a diffusion term, with the knowledge that it still converges to the transport between distributions. Moreover, as the diffusion goes to zero, the infimum converges to the $W_2$.

**Manifold Learning**    A useful assumption in representation learning is that high dimensional data originates from an intrinsic low dimensional manifold that is mapped via nonlinear functions to observable high dimensional measurements; this is commonly referred to as the manifold assumption. Formally, let $\mathcal{M}$ be a hidden $m$ dimensional manifold that is only observable via a collection of $k \gg m$ nonlinear functions $f_1, \ldots, f_k : \mathcal{M} \to \mathbb{R}$ that enable its immersion in a high dimensional ambient space as $F(\mathcal{M}) = \{\mathbf{f}(z) = (f_1(z), \ldots, f_k(z))^T : z \in \mathcal{M}\} \subseteq \mathbb{R}^k$ from which data is collected. Conversely, given a dataset $\mathsf{X} = \{x_1, \ldots, x_n\} \subset \mathbb{R}^k$ of high dimensional observations, manifold learning methods assume data points originate from a sampling $Z = \{z_i\}_{i=1}^n \in \mathcal{M}$ of the underlying manifold via $x_i = \mathbf{f}(z_i)$, $i = 1, \ldots, n$, and aim to learn a low dimensional intrinsic representation that approximates the manifold geometry of $\mathcal{M}$.

To learn a manifold geometry from collected data that is robust to sampling density variations, Coifman and Lafon [9] proposed to use an anisotropic kernel $k_{\epsilon, \beta}(x, y) := k_\epsilon(x, y)/\|k_\epsilon(x, \cdot)\|_1^\beta \|k_\epsilon(y, \cdot)\|_1^\beta$, where $0 \le \beta \le 1$ controls the separation of geometry from density, with $\beta = 0$ yielding the isotropic kernel, and $\beta = 1$ completely removing density and providing a geometric equivalent to uniform sampling of the underlying manifold. Next, the kernel $k_{\epsilon, \beta}$ is normalized to define transition probabilities $p_{\epsilon, \beta}(x, y) := k_{\epsilon, \beta}(x, y)/\|k_{\epsilon, \beta}(x, \cdot)\|_1$ and an $n \times n$ row stochastic matrix $(\mathbf{P}_{\epsilon, \beta})_{ij} := p_{\epsilon, \beta}(x_i, x_j)$ that describes a Markovian diffusion process over the intrinsic geometry of the data. Finally, a diffusion map [9] $\Phi_t(x_i)$ is defined by the eigenvalues and eigenvectors of the matrix $\mathbf{P}_{\epsilon, \beta}^t$. Most notably, this embedding preserves the diffusion distance $\|p_{\epsilon, \beta}^t(x_i, \cdot) - p_{\epsilon, \beta}^t(x_j, \cdot)/\phi_1(\cdot)\|_2$ between $x_i, x_j \in \mathsf{X}$, where $t$ is a time-scale parameter. Next, we consider a multiscale diffusion distance that relates to the geodesic distance on the manifold.

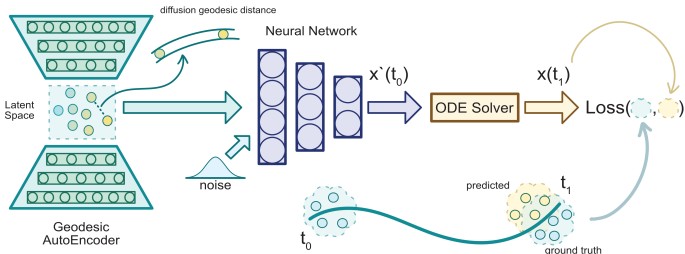

Figure 1: Overview of the MIOFlow pipeline. The Geodesic Autoencoder learns a latent space that preserves the diffusion geodesic distance. A neural network predicts the derivative of the trajectories with respect to time. For an initial sample at $t_0$, the ODE Solver produces the trajectories $x(t_1)$. At a population level, predictions of points at time $t_1$ are penalized by the Wasserstein distance during training.

**Manifold Geodesics** The dynamic optimal transport formulations (1) and (2) are valid when the ground distance is Euclidean. This is restrictive as we might have observations sampled from a lower dimensional manifold. Thus, using the Euclidean distance in the ambient space ignores the underlying geometry of the data. Static optimal transport with a geodesic distance is known to perform well on geometric domains [40, 45], or when the data lie on a manifold [46]. Here, we extend the use of a ground geodesic distance to the dynamic formulation. To do so, we define a manifold learning step that precedes the trajectory inference. The goal is to learn an embedding $\mathcal{Z}$ such that the Euclidean in $\mathcal{Z}$ is equivalent to the geodesic distance. In that case, dynamic optimal transport in $\mathcal{Z}$ is equivalent to the Wasserstein with a geodesic ground distance. In the following section, we present theoretical results that will justify our approximation of a geodesic distance on a closed Riemannian manifold.

We consider a closed Riemannian manifold $(\mathcal{M}, d_{\mathcal{M}})$, where $d_{\mathcal{M}}$ is the geodesic distance representing the shortest path between two points on the manifold. We note $h_t$ the heat kernel on $\mathcal{M}$. For a point $x \in \mathcal{M}$, the heat kernel $h_t(x, \cdot)$ induces a measure, i.e. how the heat has propagated on the manifold at time $t$ given an arbitrary initial distribution. The diffusion ground distance between $x, y \in \mathcal{M}$ is based on the $L^1$ norm between the measures induced by the heat kernel given the initial conditions $\delta_x$ and $\delta_y$.

**Definition 1.** The diffusion ground distance between $x, y \in \mathcal{M}$ is

$$D_\alpha(x, y) := \sum_{k \geq 0} 2^{-k\alpha} ||h_{2^{-k}}(x, \cdot) - h_{2^{-k}}(y, \cdot)||_1,$$

for $\alpha \in (0, 1/2)$, the scale parameter $k \geq 0$, and $h_t$ the heat kernel on $\mathcal{M}$.

Next, we state an important result from Leeb and Coifman (Thm. 2 in [25]). This theorem links the diffusion ground distance and the geodesic on a Riemannian manifold.

**Theorem 1** ([25] Thm. 2). *Let $(\mathcal{M}, d_{\mathcal{M}})$ a closed Riemannian manifold, with geodesic $d_{\mathcal{M}}$, for $\alpha \in (0, 1/2)$, the distance $D_\alpha$ is equivalent to $d_{\mathcal{M}}^{2\alpha}$.*

In practice, we cannot always evaluate the heat kernel for an arbitrary manifold. However, convergence results from Coifman and Lafon [9] provide a natural approximation. Indeed, for $\beta = 1$, the diffusion operator $P_{\epsilon, \beta}^{t/\epsilon}$ converges to the heat operator as $\epsilon$ goes to zero (Prop. 3 [9]). For the rest of the paper we assume $\beta = 1$, we define $P_\epsilon := P_{\epsilon, 1}$ and the matrix $\mathbf{P}_\epsilon := \mathbf{P}_{\epsilon, 1}$.

## 3 Manifold Interpolating Optimal-Transport Flow

To learn individual trajectories from multiple cross-sectional samples of a population, we propose MIOFlow shown in Fig. 1. Our method consists of two steps. We first learn an embedding $\mathcal{Z}$ which preserves the diffusion geodesic distance—which we define using our Geodesic Autoencoder. Then, we learn continuous trajectories based on an optimal transport loss. We model the trajectories with a Neural ODE [7], allowing us to learn non-linear paths and interpolate between timepoints.

## 3.1 Geodesic Autoencoder Embedding

In light of Thm. 1, we define an approximation of the diffusion ground distance $D_\alpha$ (Def. 1) from a collection of samples $\mathsf{X}$ that we call *diffusion geodesic distance*. We then train an autoencoder regularized to match this distance. Similar ideas are explored in [30, 11] where the latent space of the autoencoder is regularized to match a manifold learning embedding, either PHATE or diffusion map.

Our approximation relies on the diffusion matrix $\mathbf{P}_\epsilon$. We first recall its construction. We build a graph with the affinity matrix defined by $(\mathbf{K}_\epsilon)_{ij} := k_\epsilon(x_i, x_j)$, and we consider the density normalized matrix $\mathbf{M}_\epsilon := \mathbf{Q}^{-1}\mathbf{K}_\epsilon\mathbf{Q}^{-1}$, where $\mathbf{Q}$ is a diagonal matrix with $\mathbf{Q}_{ii} := \sum_j (\mathbf{K}_\epsilon)_{ij}$. In practice, we choose $k_\epsilon$ to be a Gaussian kernel or the $\alpha$-decay kernel [32]. Lastly, a Markov diffusion operator is defined by $\mathbf{P}_\epsilon := \mathbf{D}^{-1}\mathbf{M}_\epsilon$, where $\mathbf{D}_{ii} := \sum_{j=1}^n (\mathbf{M}_\epsilon)_{ij}$ is a diagonal matrix. The stationary distribution associated to $\mathbf{P}_\epsilon$ is $\boldsymbol{\pi}$, where $\boldsymbol{\pi}_i = \mathbf{D}_{ii}/\sum_j \mathbf{D}_{jj}$, since $\mathbf{P}_\epsilon$ is $\boldsymbol{\pi}$-reversible. Finally, we note that the matrix $\mathbf{P}_\epsilon$ follows a similar construction as the kernel $p_\epsilon$; the integrals are approximated by sums. We refer to [9] for more details about the convergence of the matrix operator $\mathbf{P}_\epsilon$ to the operator $P_\epsilon$. Until now, we have defined a matrix $\mathbf{P}_\epsilon$ that approximates the operator $P_\epsilon$, which in turn converges to the heat operator. Now, we define an approximation of the diffusion ground distance, based on the matrix $\mathbf{P}_\epsilon$. We use the notation $(\mathbf{P}_\epsilon)_{i:}^t$ to represent the i-th row of $\mathbf{P}_\epsilon^t$, it represents the transition probabilities of a t-steps random walk started at $x_i$.

**Definition 2.** We define the *diffusion geodesic distance* between $x_i, x_j \in \mathsf{X}$ as

$$G_\alpha(x_i, x_j) := \sum_{k=0}^{K} 2^{-(K-k)\alpha}||(\mathbf{P}_\epsilon)_{i:}^{2^k} - (\mathbf{P}_\epsilon)_{j:}^{2^k}||_1 + 2^{-(K+1)/2}||\boldsymbol{\pi}_i - \boldsymbol{\pi}_j||_1.$$

The diffusion geodesic compares the transitions probabilities of a random walk at various scales given two different initial states $x_i, x_j$. In [46], the authors use the diffusion operator $\mathbf{P}_\epsilon$ to define a distance equivalent to the Wasserstein with ground distance $D_\alpha$. Their method comes from the static formulation of optimal transport. Here we propose to learn a space $\mathcal{Z}$ that preserves an approximation of the distance $D_\alpha$, in order to do dynamic optimal transport in $\mathcal{Z}$.

**Training** To use the diffusion geodesic distance, we train a *Geodesic Autoencoder*, with encoder outputting $\phi : \mathbb{R}^k \to \mathcal{Z}$, such that $\|\phi(x_i) - \phi(x_j)\|_2^2 \approx G_\alpha(x_i, x_j)$. We draw a subsample of size $N$, evaluate $G_\alpha$ and minimize the Mean Square Error (MSE)

$$L(\phi) := \frac{2}{N} \sum_{i=1}^{N} \sum_{j>i} \left(||\phi(x_i) - \phi(x_j)||_2 - G_\alpha(x_i, x_j)\right)^2.$$

Learning the embedding $\mathcal{Z}$ over using $G_\alpha$ as the advantage of being inductive, it is useful since we use it to compute distances between predicted and ground truth observations. Moreover, the encoder can be trained to denoise, hence becoming robust to predicted values that are not close to the manifold. Computing $G_\alpha$ on the entire dataset is inefficient due to the powering of the diffusion matrix. We circumvent this difficulty with the encoder, since we train on subsamples. We choose $N$ to have few observations in most regions of the manifold, thus making the computation of $G_\alpha$ very efficient, which allows us to consider more scales.

**Theorem 2.** *Assuming $\mathsf{X}$ is sampled from a closed Riemannian manifold $\mathcal{M}$ with geodesic $d_\mathcal{M}$. Then, for all $\alpha \in (0, 1/2)$, sufficiently large $K, N$ and small $\epsilon > 0$, we have with high probability $G_\alpha(x_i, x_j) \simeq d_\mathcal{M}^{2\alpha}(x_i, x_j)$ for all $x_i, x_j \in \mathsf{X}$.*

**Corollary 1.** *If the encoder is such that $L(\phi) = 0$, then with high probability $\|\phi(x_i) - \phi(x_j)\|_2 \simeq d_\mathcal{M}^{2\alpha}(x_i, x_j)$ for all $x_i, x_j \in \mathsf{X}$.*

The stochasticity in the two previous results arises from the discrete approximation of the operator $P_\epsilon$. The law or large numbers guarantees the convergence. For fix sample size, there exist approximation error bounds with high probability, see for example [8, 39]. In practice, we choose $\alpha$ close to $1/2$ so that the diffusion distance is equivalent to the geodesic on the manifold. From that embedding $\mathcal{Z}$, we can also train a *decoder* $\phi^{-1} : \mathcal{Z} \to \mathbb{R}^k$, with a reconstruction loss $L_r := \sum_x \|\phi^{-1} \circ \phi(x) - x\|_2$. This is particularly interesting in high dimensions, since we can learn the trajectories in the lower dimensional space $\mathcal{Z}$, then decode them in the ambient space. For example, we use the decoder in

Sec. 4 to infer cellular trajectories in the gene space, enabling us to understand how specific genes evolved over time. We describe the training procedure of the GAE in algorithm 2 in the supplementary material.

## 3.2 Inferring Trajectories

Given $T$ distributions $\{\mu_i\}_{i=0}^{T-1}$, we want to model the trajectories respecting the conditions $X_i \sim \mu_i$ for fix timepoints $i \in \{0, \ldots, T-1\}$. Formally, we want to learn a parametrized function $f_\theta(x,t)$ such that

$$X_t = X_0 + \int_0^t f_\theta(X_u, u)du, \text{ with } X_0 \sim \mu_0, \ldots, X_{T-1} \sim \mu_{T-1}. \tag{4}$$

We adapt the main theorem from [44], to consider the path space, and other types of dissimilarities between distributions.

**Theorem 3.** *We consider a time-varying vector field $f(x,t)$ defining the trajectories $dX_t = f(X_t, t)dt$ with density $\rho_t$, and a dissimilarity between distributions such that $D(\mu, \nu) = 0$ iff $\mu = \nu$. Given these assumptions, there exist a sufficiently large $\lambda > 0$ such that*

$$W_2(\mu, \nu)^2 = \inf_{X_t} E\left[\int_0^1 \|f(X_t, t)\|_2^2 dt\right] + \lambda D(\rho_1, \nu) \text{ s.t. } X_0 \sim \mu. \tag{5}$$

*Moreover, if $X_t$ is defined on the embedded space $\mathcal{Z}$, then $W_2$ is equivalent to the Wasserstein with geodesic distance $W_2(\mu, \nu) \simeq W_{d_{\mathcal{M}}^{2\alpha}}(\mu, \nu)$.*

This theorem enables us to add the second marginal constraint in the optimization problem. In practice, it justifies the method of matching the marginals and adding a regularization on the vector field $f(x,t)$, that way the optimal $(\rho_t, f)$ corresponds to the one from the $W_2$. If in addition we model the trajectories in the embed space $\mathcal{Z}$, then the transport is equivalent to the one on the manifold.

**Training** We observe discrete distributions $\boldsymbol{\mu}_i := (1/n_i)\sum_i \delta_{x_i}$ for $x_i \in \mathsf{X}_i$, and we approximate (4) with a Neural ODE [7], where $f_\theta$ is modeled by a neural network with parameters $\theta$. Denote by $\psi_\theta : \mathbb{R}^k \times \mathcal{T} \to \mathbb{R}^{d|\mathcal{T}|}$ the function that represents the Neural ODE, where $\mathcal{T}$ is a set of time indices. We define the predicted sets $\widehat{\mathsf{X}}_1, \ldots, \widehat{\mathsf{X}}_{T-1} = \psi_\theta(\mathsf{X}_0, \{1, \ldots, T-1\})$; given an initial set $\mathsf{X}_0$ it returns the approximation of (4) for all $t \in \mathcal{T}$. The resulting discrete distributions are $\hat{\boldsymbol{\mu}}_i := (1/n_i)\sum_j \delta_{x_j}$ for $x_j \in \widehat{\mathsf{X}}_i$. To match the marginals, we used two approaches of training; *local* or *global*. For the local method, we only predict the next sample, hence given $\mathsf{X}_t$ as initial condition, we predict $\widehat{\mathsf{X}}_{t+1} = \psi_\theta(\mathsf{X}_t, t+1)$. Whereas for the global, we use the entire trajectories given the initial condition $\mathsf{X}_0$. For both cases, we formulate a loss $L_m$ on the marginals, representing the second term in (5). To take into account the first term in (5), we add the loss $L_e$, where the integral can be approximate with the forward pass of the ODE solver, and $\lambda_e \geq 0$ is a hyperparameter.

$$L_m := \sum_{i=1}^{T-1} W_2(\hat{\boldsymbol{\mu}}_i, \boldsymbol{\mu}_i) \qquad L_e := \lambda_e \sum_{i=1}^{T-1} \int_{i-1}^{i} \|f_\theta(x_t, t)\|_2^2 dt \tag{6}$$

In practice, to compute the Wasserstein between discrete distributions, we use the implementation from the library Python Optimal Transport [14]. This is in contrasts with the maximum-likelihood approach in CNF methods which requires evaluation of the instantaneous change of variables at every integration timestep at $O(k^2)$ additional cost per function evaluation [7].

We add a final density loss $L_d$ inspired by [44] to encourage the trajectories to stay on the manifold. Intuitively, for a predicted point $x \in \widehat{\mathsf{X}}_t$, we minimize the distance to its k-nearest neighbors in $\mathsf{X}_t$ given a lower bound $h > 0$

$$L_d := \lambda_d \sum_{t=1}^{T-1} \sum_{x \in \widehat{\mathsf{X}}_t} \ell_d(x, t), \text{ where } \ell_d(x, t) := \sum_{i=1}^{k} \max(0, \text{min-k}(\{\|x - y\| : y \in \mathsf{X}_t\}) - h).$$

We describe the overall training procedure of MIOFlow in algorithm 1.

**Modeling Diffusion**   We add a diffusion term in the trajectories, resulting in the SDE $dX_t = f(X_t, t)dt + \sqrt{\sigma_t}dB_t$ In practice, we learn $T$ parameters $\{\sigma_t\}_{t=0}^{T-1}$ It has the advantage of mapping the same cell to different trajectories stochastically as in many naturalistic systems. Early in the training process, we notice that it promotes bifurcation, as it helps to spread the initial $X_0$. As training continues, we empirically observed that $\sigma \to 0$. Hence, as shown in [28], converging to the $W_2$ formulation. For small $\sigma_t$, we train with a regular ODE solver, essentially corresponding to the Euler–Maruyama method. To model complex diffusion, one would need an SDE solver, thus drastically increasing the computational cost.

---

**Algorithm 1** MIOFlow

---

**Input:** Datasets $X_0, \ldots, X_{T-1}$, graph kernel $\mathbf{K}_\epsilon$, maximum scale $K$, noise scale $\xi$, batch size $N$, maximum iteration $n_{max}$, initialized encoder $\phi$ and decoder $\phi^{-1}$, maximum local iteration $n_{local}$, maximum global iteration $n_{global}$, initialized neural ODE $\psi_\theta$.
**Output:** Trained neural ODE $\psi_\theta$.
**GAE:** $\phi, \phi^{-1} \leftarrow \text{GAE}(X, \mathbf{K}_\epsilon, K, \xi, N, n_{max}, \phi, \phi^{-1})$         ▷ See algorithm 2 in SM
**if** use GAE **then** $X_0, \ldots, X_{T-1} \leftarrow \phi(X_0), \ldots, \phi(X_{T-1})$
**end if**
**for** i=1 **to** $n_{local}$ **do**                                               ▷ Local training
    **for** t=0 **to** $T-2$ **do**
        $\widehat{X}_{t+1} \leftarrow \psi_\theta(X_t, t+1)$
    **end for**
    $L \leftarrow L_m + L_e + L_d$
    update $\psi_\theta$ with gradient descent w.r.t. the loss $L$
**end for**
**for** i=1 **to** $n_{global}$ **do**                                       ▷ Global training
    $\widehat{X}_1, \ldots, \widehat{X}_{T-1} \leftarrow \psi_\theta(X_0, \{1, \ldots, T-1\})$
    $L \leftarrow L_m + L_e + L_d$
    update $\psi_\theta$ with gradient descent w.r.t. the loss $L$
**end for**

---

## 4   Results

Here, we validate our method empirically on both synthetic test cases and single-cell RNA sequencing datasets chosen to test the capability of MIOFlow to follow divergent trajectories from population snapshot data. Using these test cases we assess whether MIOFlow-derived trajectories qualitatively follow the data manifold efficiently, we also quantitatively assess their accuracy using the Wasserstein $W_1$, and Maximum Mean Discrepancy (MMD) [18] on held-out timepoints. The MMD is evaluated with the Gaussian (G) kernel and the mean (M). We compare our results to those other methods that preform population flows including TrajectoryNet [44] which is based on a CNF that is regularized to achieve efficient paths, and Diffusion Schrödinger's Bridge (DSB) [10] which is an optimal transport framework for generative modeling. The baseline measure in the quantitative results corresponds to the average distance between the previous and next timepoints for a given time $t$. Additional experiments, such as ablation studies, and details are provided in the supplementary material.

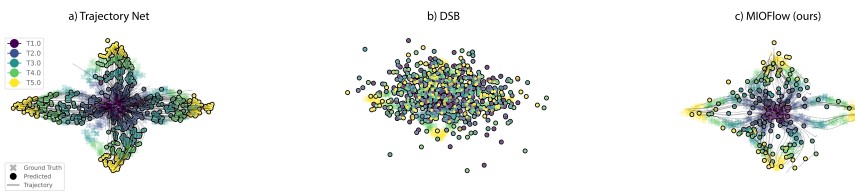

Figure 2: Comparisons between predicted flows of (a) TrajectoryNet, (b) DSB, and (c) MIOFlow on a toy Petal dataset. MIOFlow trajectories both match the data and transition along the data manifold whereas TrajectoryNet takes shortcut paths between the the curved paths and Schroedinger's bridge does not follow the data. Time is indicated by color. Ground truth points are designated by translucent "X"s in the background, whereas predicted points are opaque circles, whilst trajectories are solid lines.

## 4.1 Artificial Data

**Petal** The petal dataset is a simple yet challenging, as it mimics natural dynamics arising in cellular differentiation, including bifurcations and merges. The dynamics start in the center (purple) and evolves through the end of the leaf (yellow). In Fig. 2, we present the trajectories on the petal dataset for TrajectoryNet, DSB, and MIOFlow. Only our method is able to learn the dynamic, and keeping trajectories on the manifold. We then hold-out timepoint 2 corresponding to the initial bifurcation, and, in Tab. 1, we compare the interpolation between the three methods. MIOFlow results in a more accurate entropic interpolation for both $W_1$ and MMD(G), while requiring much less computational time. In the supplementary material, we present additional results for DSB on the petal dataset.

**Dyngen** We use Dyngen [6] to simulate a scRNA-seq dataset from a dynamical cellular process. We first embed the observations in five dimensions using the non-linear dimensionality reduction method PHATE [32]. We discretize the space in five bins, each representing a differentiation stage. This dataset includes one bifurcation. However, it proves more challenging than the petal dataset. The number of samples per bins is not equal, which tends to give more influence to the lower branch. Moreover, the bifurcation is asymmetric; the curve in the lower branch is less pronounce than the upper one, making it harder to learn a continuous bifurcation. In Fig. 3, we show the trajectories for TrajectoryNet, DSB, and our method. MIOFlow is more accurate as it learns continuous bifurcating trajectories along the two main branches. In Tab. 1, we hold-out timepoint 2, and show the interpolation results against the ground truth. MIOFlow is more accurate for all metrics, while requiring much less computational time. We see that in timepoint 1, TrajectoryNet deviates far from the trajectories particularly at the branching point where it struggles with induced heterogeneity (see Fig. 3). We have further quantifications reflecting this in the supplementary material.

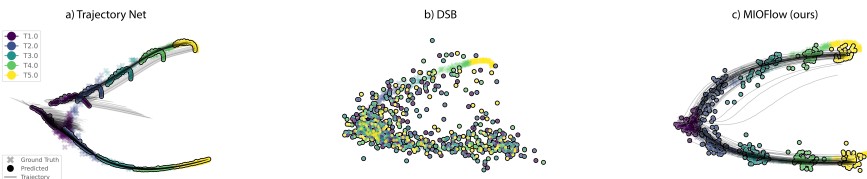

Figure 3: Comparisons between predicted flows of (a) TrajectoryNet, (b) DSB, and (c) MIOFlow on Dyngen [6] simulated dataset. MIOFlow trajectories both match the data and transition along the data manifold throughout the timepoints. Time is indicated by color. Ground truth points are designated by translucent "X"s in the background, whereas predicted points are opaque circles, whilst trajectories are solid lines.

Table 1: Leave-one-out $W_1$ and MMD with (G)aussian or (M)ean kernel between the predicted and ground truth points as well as training time in (m)inutes and (s)econds on the Petal and Dyngen datasets of TrajectoryNet, DSB, and MIOFlow. Lower is better.

|  | Petal | | | | Dyngen | | | |
|  | $W_1$ | MMD(G) | MMD(M) | Runtime | $W_1$ | MMD(G) | MMD(M) | Runtime |
|---|---|---|---|---|---|---|---|---|
| MIOFlow (ours) | **0.090** | **0.029** | 0.005 | **284.60** (s) | **0.783** | **0.199** | **0.509** | **95.63** (s) |
| TrajectoryNet | 0.181 | 0.136 | 0.002 | 64 (m) | 1.499 | 0.303 | 1.806 | 62 (m) |
| DSB | 0.199 | 0.157 | 0.008 | 86 (m) | 2.051 | 0.932 | 2.367 | 81 (m) |
| Baseline | 0.221 | 0.196 | $<$**0.001** | N/A | 1.388 | 1.008 | 0.926 | N/A |

## 4.2 Single-Cell Data

**Embryoid Body** Here we evaluate our method on scRNA-seq data from the dynamic time course of human embryoid body (EB) differentiation over a period of 27 days, measured in 3 timepoints between days 0-3, 6-9, 12-15, 18-21, and 24-27. We reduce the dimensionality of this dataset to 200 dimensions using PCA. See the supplementary material for further details on the preprocessing steps and the parameters.

By combining PCA and the autoencoder with diffusion geodesic distance described in section 3.1, we can decode the trajectories back into the gene space (17846 dimensions), thus describing the evolution of the gene expression of a cell. In Fig. 5 we plot the trajectories, at a gene level of 20 neuronal

cells against 20 random one. We generally observe distinct differences between TrajectoryNet and MIOFlow. TrajectoryNet (which is shown in the bottom) shows a tendency to have early divergence in trends artificially showing higher heterogeneity in gene trends then exist in neuronal specification processes. Examples include HAND2, which has a non-monotonic trend, first increasing in expression during development from embryonic stem cells to neural early progenitors and then decreasing as the lineages mature [26]. MIOFlow clearly shows this trend whereas TrajectoryNet trends are more chaotic. Another example is ONECUT2 which is consistently known to be higher in cells that diverge towards the neuronal lineage [48]. Other examples are discussed in the supplementary material.

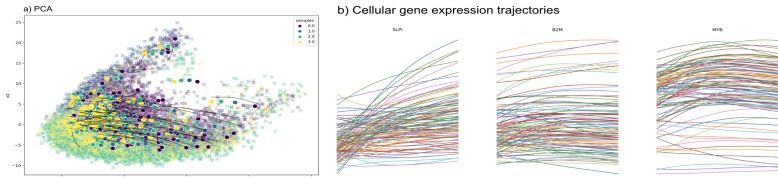

Figure 4: MIOFlow on the AML dataset. a) PCA Embedding comparing ground truth to MIOFlow predicted points and trajectories. b) Cellular gene expression trajectories for *SLPI*, *B2M*, and *MYB* for MIOFlow.

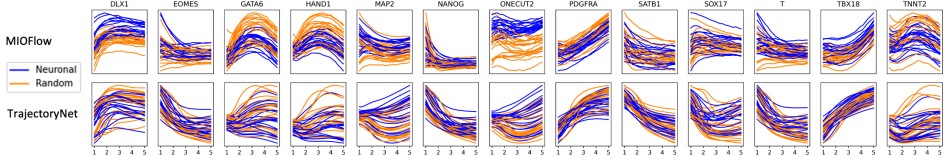

Figure 5: Neuronal gene trajectories vs. random trajectories for MIOFlow (top) vs. TrajectoryNet [46] (bottom). TrajectoryNet consistently shows a tendency for early divergence in trajectories, examples include ONECUT2 which is biologically known to be higher expressed [32] and HAND2 which is known to peak in neural early progenitors (NEPs).

In Tab. 2, we compare the prediction accuracy of MIOFlow whether we use the GAE embedding or no embedding. For the GAE, we show results for two kernels; the Gaussian kernel (scale $0.5$), and the $\alpha$-decay kernel [32] (based on a 30 nearest neighbors graph). For almost all metrics, the prediction accuracy is greater with the GAE. We present additional results for the choice of kernel in the supplementary material.

Table 2: Leave-one-out $(t)$ $W_1$ and MMD with (G)aussian and (M)ean kernel between the predicted and ground truth point on the EB datasets for MIOFlow with the Gaussian, $\alpha$-decay kernel, or without GAE. Lower is better.

|  | $t = 2$ | | | $t = 3$ | | |
|---|---|---|---|---|---|---|
|  | $W_1$ | MMD(G) | MMD(M) | $W_1$ | MMD(G) | MMD(M) |
| GAE Gaussian | 26.318 | **0.059** | 101.969 | 32.963 | 0.138 | 223.679 |
| GAE $\alpha$-decay | **25.744** | 0.061 | **99.747** | **32.227** | 0.135 | **213.465** |
| No GAE | 29.243 | 0.063 | 126.608 | 35.709 | 0.119 | 246.609 |
| Baseline | 33.415 | 0.103 | 227.279 | 35.319 | **0.095** | 213.492 |

**AML Cancer Data with Treatment-based Progression** We also assessed whether MIOFlow can give insights on treatment resistant cancer cell populations, rather than differentiation. For this purpose, we used data from Fennell et al. [12] in which AF9 mutant cancer cells implanted into the bone marrow of mice as a model of Acute Myeloid Leukemia (AML). The mice are then treated for 7 days with intensive chemotherapy with single cell measurements taken at day 2, day 5, and day 7. Here we show MIOFlow on this dataset along with signatures of leukemic stem cells (LSCs) which are hypothesized to be dominant in chemotherapy resistant cells. We see heterogeneity in the trajectories, the majority of cells surviving to the last timepoint showing an increase in the markers SLPI, MYB, and B2M—all members of the LSC pathway [12]. This indicates that not all cells start with this signature but are made to transition to this signature to resist the effects of chemotherapy.

# 5   Related Work

The task of recovering dynamics from multiple snapshots has gained a lot of attention, especially in biology with scRNA-seq data. Static methods such as pseudotime assign an abstract measure of time for all cells depending on an initial state [19, 47]. In can be combined to infer the cell lineage, i.e. the principal curves or trajectories [43]. When estimates of cell growth rates are available, Zhang et al. [52] proposed a method based on optimal transport. Another line of research is on the direct rate of differentiation of cell or RNA velocity [23, 4]. This method does not provide trajectories, but it can be combined to define a transition matrix to infer trajectories [53]. Saelens et al. [37] provide a rigorous comparison of 45 static methods.

Similar methods used the dynamic formulation of optimal transport [5, 13, 20, 35, 36, 38, 44] to infer trajectories. The dynamics are either modeled with a recurrent neural network [20], a generative adversarial networks [35], flows defined by the gradient of an input convex neural network [5], or a CNF [44]. Here, we circumvent the need for a CNF by modeling the trajectories directly on the support of the distributions, hence maintaining the computational cost low even in high dimension. Some generative methods that interpolate between a prior to a target can also be used to recover trajectories. Other methods use the Schrödinger bridge (SB) [10, 22, 49, 51] problem to define trajectories between distributions. For example [10], solve the SB between two distributions using a series of Sinkhorn iterations. However, it does not explicitly learn dynamics, making their interpolations inaccurate.

# 6   Conclusion

We proposed MIOFlow a computationally efficient method to recover trajectories from multiple cross-sectional samples of a time-evolving population. Our method includes a novel manifold learning framework, where we learn an embedding preserving the geodesic distance. We show theoretically how MIOFlow is related to dynamical optimal transport on an underlying manifold, and it's capacity to model stochastic trajectories. On artificial datasets, MIOFlow produced continuous trajectories more in line with the manifold compared to DSB and TrajectoryNet. On scRNA-seq data, we leveraged an autoencoder with the geodesic embedding to learn the trajectories in a lower dimensional space while preserving meaningful distances which can then be decoded in gene space. This way we could visualize the evolution of the gene expression of a cell.

# 7   Limitations and Broader Impact

Our method does have some limitations. Similarly to other works based on continuous time networks, our work makes use of generic ODE solvers and adjoint dynamics which can fail under stiff dynamics. Like other methods that learn dynamics it is possible that the learned dynamics may become stiff through training leading to decreased model accuracy. In addition, our work is sensitive to the time-bins, and it worked best for relatively uniform sampling. This second issue could potentially be alleviated by using unbalanced optimal transport, which allows transporting distributions with unequal masses. Our work can help understand dynamical systems, and in particular cellular differentiation, and thus to the best of our knowledge presents no potential negative societal impacts.

## Acknowledgments and Disclosure of Funding

We thank Tom Marty for helping to prototype the GAE. This work was partially funded and supported by CIFAR AI Chair [G.W.], NSERC Discovery grant 03267 [G.W.], NIH grants 1F30AI157270-01 [M.K.], R01GM135929 [G.W., S.K.], R01HD100035, and R01GM130847 [G.W., S.K.], NSF Career grant 2047856 [S.K.], the Chan-Zuckerberg Initiative grants CZF2019-182702 and CZF2019-002440 [S.K.], and the Sloan Fellowship FG-2021-15883 [S.K.]. The content provided here is solely the responsibility of the authors and does not necessarily represent the official views of the funding agencies. The funders had no role in study design, data collection & analysis, decision to publish, or preparation of the manuscript.

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
