# Supplementary Material

## A  Theory and Algorithm

### A.1  Geodesic Autoencoder Algorithm

Using the definitions from Sec. 3, we present the algorithm to train the geodesic autoencoder.

---

**Algorithm 2** Geodesic Autoencoder (GAE)

---

**Input:** Dataset $\mathsf{X}$ of size $n$, graph kernel $\mathbf{K}_\epsilon$, maximum scale $K$, noise scale $\xi$, batch size $N$, maximum iteration $n_{max}$, initialized encoder $\phi$ and decoder $\phi^{-1}$.
**Output:** Trained encoder $\phi$ and decoder $\phi^{-1}$.
**for** i=1 **to** $n_{max}$ **do**
    Sample batch $\{x_1, \ldots, x_N\} \subseteq \mathsf{X}$ of size $N$
    $\mathbf{Q} \leftarrow \mathrm{diag}(\sum_j (\mathbf{K}_\epsilon)_{ij})$
    $\mathbf{M}_\epsilon \leftarrow \mathbf{Q}^{-1}\mathbf{K}_\epsilon\mathbf{Q}^{-1}$
    $\mathbf{D} \leftarrow \mathrm{diag}(\sum_j (\mathbf{M}_\epsilon)_{ij})$
    $\mathbf{P}_\epsilon \leftarrow \mathbf{D}^{-1}\mathbf{M}_\epsilon$                        ▷ $N \times N$ diffusion matrix
    $G_\alpha \leftarrow 2^{-K\alpha}\,\mathrm{pairwise\_distances}((\mathbf{P}_\epsilon)_{1:}, \ldots, (\mathbf{P}_\epsilon)_{N:})$
    **for** k=1 **to** $K$ **do**
        $G_\alpha \leftarrow G_\alpha + 2^{-(K-k)\alpha}\,\mathrm{pairwise\_distance}((\mathbf{P}_\epsilon^{2^k})_{1:}, \ldots, (\mathbf{P}_\epsilon^{2^k})_{N:})$
    **end for**
    $\tilde{x}_1, \ldots, \tilde{x}_N \leftarrow x_1 + \xi z_1, \ldots, x_N + \xi z_N$, where $\{z_1, \ldots, z_N\} \sim \mathcal{N}(0, 1)$
    $L \leftarrow \mathrm{MSE}\big(\mathrm{pairwise\_distances}(\phi(\tilde{x}_1), \ldots, \phi(\tilde{x}_N)), G_\alpha\big)$
    $L_r \leftarrow \sum_i \|\phi^{-1} \circ \phi(\tilde{x}_i) - x\|_2$
    **Update** $\phi, \phi^{-1}$ with gradient descent w.r.t. $L(\phi), L_r(\phi, \phi^{-1})$
**end for**

---

### A.2  Proof of Thm.2

**Theorem 2.** Assuming $\mathsf{X}$ is sampled from a closed Riemannian manifold $\mathcal{M}$ with geodesic $d_\mathcal{M}$, then for a sufficiently large $K, N$ and small $\epsilon > 0$, we have $G_\alpha(x_i, x_j) \simeq d_\mathcal{M}^{2\alpha}(x_i, x_j)$, for all $\alpha \in (0, 1/2)$.

*Proof.* We adapt the convergence proof from [46], it can be summarized in two parts. First, the equivalence of the distances defined by $P_\epsilon^{t/\epsilon}$ and the heat operator $H_t$. This relies on the convergence results $P_\epsilon^{t/\epsilon} \to H_t$ as $\epsilon \to 0$ from Coifman and Lafon [9]. Second, the approximation of the operator $P_\epsilon$ based on a finite set of points. Throughout the proof we use the same notation $\delta_x$ for the delta Dirac function, and $\delta_{x_i}$ for a row vector with $i$-th element 1, and 0 otherwise.

For a family of operator $(A_t)_{t \in \mathbb{R}}$ we define

$$D_{\alpha, A}(x, y) := \sum_{k \geq 0} 2^{-k\alpha}||A_{2^{-k}}(x, \cdot) - A_{2^{-k}}(y, \cdot)||_1,$$

and equivalently $D_{\alpha, A}(x, y) = \sum 2^{-k\alpha}||A_{2^{-k}}(\delta_x - \delta_y)||_1$, where $A\delta_x := \int A(u, \cdot)\delta_x(u)du$. This definition is the same as Def. 1, but for an arbitrary operator. Our final goal is to show that $G_\alpha(x_i, x_j) \simeq D_{\alpha, H_t}(x_i, x_j)$, and then conclude with Thm. 1 which gives the equivalence $D_{\alpha, H_t}(x_i, x_j) \simeq d_\mathcal{M}^{2\alpha}(x_i, x_j)$.

We first want to show that, for a sufficiently small $\epsilon > 0$, we have $D_{\alpha, P_\epsilon^{t/\epsilon}}(x, y) \simeq D_{\alpha, H_t}$, where $P$ is the anisotropic operator defined in Sec. 2. From Coifman and Lafon [9], we have $\|P_\epsilon^{t/\epsilon} - H_t\|_{L^2(\mathcal{M})} \to 0$ as $\epsilon \to 0$, and also $\|P_\epsilon^{t/\epsilon} - H_t\|_{L^1(\mathcal{M})} \to 0$.

Now let $\Gamma_t := P_\epsilon^{t/\epsilon} - H_t$, and for $\gamma > 0$ choose $\epsilon > 0$ such that $\|\Gamma_t(\delta_x - \delta_y)\|_1 < \gamma\|\delta_x - \delta_y\|_1 = 2\gamma$ for $x \neq y$, we have

$$D_{\alpha,\Gamma_t}(x,y) = \sum_{k\geq 0} 2^{-k\alpha}\|\Gamma_{2^{-k}}(\delta_x - \delta_y)\|_1 \leq 2\gamma\sum_{k\geq 0} 2^{-k\alpha} = \frac{2\gamma}{1-2^{-\alpha}}.$$

Thus, for all $t > 0$, and all $\gamma > 0$ there exists $\epsilon > 0$ such that $D_{\alpha,\Gamma_t}(x,y) < \gamma$. From the reverse triangle inequality, we have

$$|D_{\alpha,P_\epsilon^{t/\epsilon}}(x,y) - D_{\alpha,H_t}| \leq D_{\alpha,\Gamma_t}(x,y),$$

using $D_{\alpha,\Gamma_t}(x,y) < \gamma$, we obtain

$$D_{\alpha,H_t}(x,y) - \gamma \leq D_{\alpha,P_\epsilon^{t/\epsilon}}(x,y) \leq D_{\alpha,H_t}(x,y) + \gamma.$$

According to [59] we can lower bound the heat kernel, and thus the distance $D_{\alpha,H_t}(x,y) > C$ for some $C > 0$. For $\gamma < C/2$, and a sufficiently small $\epsilon > 0$, we have

$$(1/2)D_{\alpha,H_t}(x,y) \leq D_{\alpha,P_\epsilon^{t/\epsilon}}(x,y) \leq (3/2)D_{\alpha,H_t}(x,y).$$

This proves our first claim, that $D_{\alpha,P_\epsilon^{t/\epsilon}}(x,y) \simeq D_{\alpha,H_t}$ for small $\epsilon > 0$.

Next, we consider the approximation of the operator $P_\epsilon^{t/\epsilon}$ with a finite set of points. Now we define

$$G_{K,\epsilon,\alpha}(x_i,x_j) := \sum_{k=0}^{K} 2^{-k\alpha}\|(\delta_{x_i} - \delta_{x_j})\mathbf{P}_\epsilon^{2^{-k}/\epsilon}\|_1,$$

if $\epsilon = 2^{-K}$, then the first term of $G_\alpha(x_i,x_j)$ is equal to $G_{K,\epsilon,\alpha}(x_i,x_j)$. Since we will let $K \to \infty$, we ignore the second term in $G_\alpha(x_i,x_j)$, as it converges to zero. Similar to Coifman and Lafon [9] (Sec. 5), we have $\mathbf{P} \to P$ as $n \to \infty$, using Monte-Carlo integration (approximation of integral with summation by the law of large numbers). By the strong law of large numbers, the convergence is with probability one. For a finite number of samples, we have a high probability bound on the convergence, see for example [8, 39]. Now we let $N := \min(K,n)$, and therefore

$$\lim_{N\to\infty} G_{K,\epsilon,\alpha}(x_i,x_j) = D_{\alpha,P_\epsilon^{t/\epsilon}}(x_i,x_j).$$

Hence, if we take $\epsilon := 2^{-K}$, we have for sufficiently large $n$ and $K$ (implying small $\epsilon > 0$), $G_\alpha(x_i,x_j) \simeq D_{\alpha,H_t} \simeq d_{\mathcal{M}}^{2\alpha}(x_i,x_j)$ from Thm. 1. $\qquad\square$

### A.3 Proof of Thm.3

**Theorem 3.** We consider a time-varying vector field $f(x,t)$ defining the trajectories $dX_t = f(X_t,t)dt$ with density $\rho_t$, and a dissimilarity between distributions such that $D(\mu,\nu) = 0$ iff $\mu = \nu$. Given these assumptions, there exist a sufficiently large $\lambda > 0$ such that

$$W_2(\mu,\nu)^2 = \inf_{X_t} E\left[\int_0^1 \|f(X_t,t)\|_2^2 dt\right] + \lambda D(\rho_1,\nu) \text{ s.t. } X_0 \sim \mu.$$

Moreover, if $X_t$ defined on the embedded space $\mathcal{Z}$, then $W_2$ is equivalent to the Wasserstein with geodesic distance $W_2(\mu,\nu) \simeq W_{d_{\mathcal{M}}^{2\alpha}}(\mu,\nu)$.

*Proof.* We recall that

$$W_2(\mu,\nu)^2 = \inf_{X_t} \mathbb{E}\left[\int_0^1 \|f(X_t,t)\|_2^2 dt\right] \text{ s.t. } dX_t = f(X_t,t)dt, X_0 \sim \mu, X_1 \sim \nu,$$

is equivalent to

$$W_2(\mu,\nu)^2 = \inf_{(\rho_t,v)} \int_0^1 \int_{\mathbb{R}^k} \|v(x,t)\|^2 \rho_t(dx)dt,$$

with the three constraints

$$a)\, \partial_t \rho_t + \nabla \cdot (\rho_t v) = 0 \quad b)\, \rho_0 = \mu, \quad c)\, \rho_1 = \nu.$$

Tong et al. [44] (Thm. 4.1), showed that, for large $\lambda > 0$ and $\rho_t$ satisfying *a)*, this minimization problem is equivalent to

$$W_2(\mu, \nu)^2 = \inf_{(\rho_t, v)} \int_0^1 \int_{\mathbb{R}^k} \|v(x,t)\|^2 \rho_t(dx) dt + \lambda KL(\rho_1 \,\|\, \nu),$$

where KL is the Kullback–Leibler divergence, we note that their proof is valid for any dissimilarity $D(\rho_1 \,\|\, \nu)$ respecting the identity of indiscernibles. Using the path formulation, by writing the integral as an expectation and taking the infimum over all absolutely continuous path, we have

$$W_2(\mu, \nu)^2 = \inf_{X_t} E\left[ \int_0^1 \|f(X_t, t)\|_2^2 dt \right] + \lambda D(\rho_1, \nu) \text{ s.t. } dX_t = f(X_t, t)dt, \, X_0 \sim \mu.$$

Assuming the encoder $\phi$ achieves a loss of zero, then from Corr. 1 for $\alpha \in (0, 1/2)$, we have $\|\phi(x_i) - \phi(x_j)\|_2 \simeq d_{\mathcal{M}}^{2\alpha}(x_i, x_j)$ for all $x_i, x_j \in \mathsf{X} \subseteq \mathcal{M}$, and sufficiently large $n$ and $K$. That is, there exist $c, C > 0$ such that $c\, d_{\mathcal{M}}(x_i, x_j) \leq \|\phi(x_i) - \phi(x_j)\|_2 \leq C d_{\mathcal{M}}(x_i, x_j)$ for all $x_i, x_j \in \mathsf{X}$. Then, for all $\pi \in \Pi(\mu, \nu)$, we have

$$c^p \int_{\mathsf{X} \times \mathsf{X}} d_{\mathcal{M}}^{2\alpha}(x, y)^p \pi(dx, dy) \leq \int_{\mathsf{X} \times \mathsf{X}} \|\phi(x) - \phi(y)\|_2^p \pi(dx, dy) \leq C^p \int_{\mathsf{X} \times \mathsf{X}} d_{\mathcal{M}}^{2\alpha}(x, y)^p \pi(dx, dy),$$

and taking the infimum with respect to $\pi \in \Pi(\mu, \nu)$ yields the desired result $W_2(\mu, \nu) \simeq W_{d_{\mathcal{M}}^{2\alpha}}(\mu, \nu)$.

□

### A.4 Schrödinger bridge

Let $\mathcal{D}(\mu, \nu)$ the space of probability distributions on $\mathcal{C} := C([0, 1], \mathbb{R}^k)$, with initial and final distribution $\mu$ and $\nu$, and $N$ the Wiener measure on $\mathcal{C}$, then the Schrödinger bridge (SB) problem is to find the time-evolving distribution $\rho_t$ such that

$$\min KL(\rho_t \,\|\, N) \text{ subject to } \rho_t \in \mathcal{D}(\mu, \nu),$$

where KL is the Kullback–Leibler divergence. It also admits a static formulation, i.e. minimizing with respect to $\pi \in \Pi(\mu, \nu)$. Using the measure $N_x^y$ over a Brownian bridge with initial and final condition $x$ and $y$, and $\rho_x^y$ defined similarly for $\rho \in \mathcal{D}(\mu, \nu)$, we can write $KL(\rho_t \,\|\, N) = KL(\pi \,\|\, \pi^N) + KL(\rho_x^y \,\|\, N_x^y)$, where $\pi^N$ is the joint distribution between the initial and final states under the Wiener measure $N$. By choosing $\rho_x^y = N_x^y$, the *static formulation* of the Schrödinger bridge is

$$\min KL(\pi \,\|\, \pi^N) \text{ subject to } \pi \in \Pi(\mu, \nu).$$

When the Wiener measure has variance $\sigma^2$, the static Schrödinger bridge is equivalent to minimizing

$$\inf_{\pi \in \Pi(\mu, \nu)} \int_{\mathbb{R}^k \times \mathbb{R}^k} (1/2) \|x - y\|_2^2 \pi(dx, dy) + \sigma^2 H(\pi),$$

where $H$ is the entropy (see [33] and references therein). That is an optimal transport problem with entropic regularization, the same formulation as the Sinkhorn divergence [54], which can be solved with Sinkhorn's algorithm [58]. We compare our method with Diffusion Schrödinger Bridge (DSB) [10], which solve the SB problem with an approximation of the iterative proportional fitting algorithm.

## B  Experiment details

For all experiments, we used the *Runge-Kutta* RK4 ODE solver, for the density loss we used $h = 0.01$, and 5 nearest neighbors. The ODE network consists of three layers, and we concatenate two extra dimensions to the input as well as the time index. The encoder network is three layers with ReLU activation functions in between layers. For each epoch (local or global), we sampled 20 batches of a given sample size without replacement. For both networks, we optimize with AdamW [57] with default parameters. In practice, we did not use the energy loss, as our trajectories were already smooth, and it requires more careful parametrization (see Sec. C). For all experiments, we used TrajectoryNet with 1000 iterations and *whiten*. We used the authors' implementation of DSB with the *basic* model and default parameters.

**Petal** For diffusion geodesic, we experiment with an RBF kernel with scale $0.1$ and $\alpha$-decay kernel (knn=5), both with maximum time scale $2^5$. The encoder consists of two linear layers of size 8 and 32, trained for 1000 iterations. We used LeakyReLU in between layers of the ODE network, with hidden layers size $16, 32, 16$, and initial scales $\sigma_t = 0.1$. We trained for 40 local epoch with sample size 60. We used $\lambda_d = 35$ to weight the density loss and $\lambda_e = 0.001$. For the hold-out experiments, we trained for 30 local and 15 global epoch with $\lambda_d = 1$.

The diffusion parameters are generally small while training, for example when trained on the entire dataset $\sigma_t \in \{0.18, 0.15, -0.03, 0.11, 0.08\}$, and for the hold-out time three $\sigma_t \in \{0.08, 0.02, 0.16, 0.03, 0.08\}$.

**Dyngen** The diffusion geodesic was evaluated with an RBF kernel with scale $0.5$, and maximum time scale $2^5$. The encoder has two linear layers of size 8 and 32, trained for 1000 iterations. We used CELU in between layers of the ODE network, with hidden layers size $16, 32, 16$, and initial scales $\sigma_t = 0.2$. We trained for 50 global epochs of size 60, with $\lambda_d = 5$. For the hold-out experiments, we used $\lambda_d = 5$, and for $t = 5$ we trained on 5 local and 10 global epochs. For $t = 3$, since we have to hold-out the steps $(2, 3)$ and $(3, 4)$, we only trained using 50 global epochs.

Similar to the petal dataset, the diffusion parameters is empirically low, on the entire dataset it is $\sigma_t \in \{0.18, 0.20, 0.23, 0.18, 0.16\}$, and for the hold-out three $\sigma_t \in \{0.02, 0.08, 0.24, 0.22, 0.24\}$.

**EB data** The EB data are publicly available[4]. For the genes trajectories we used the full geodesic autoencoder because we want to decode in PCA space, and then project back into gene space. The encoder consists of three layers of size $200, 100, 100$, and the decoder is $100, 100, 200$, both with ReLU activation between layers. We trained for 1000 epochs with sample size of 100 per time, with the $\alpha$-decay kernel (5 nearest neighbors, and max scale of $2^6$). This kernel is more expensive to compute since it relies on the approximation of the knn, but it circumvents the need to find a good scale as with the RBF kernel. The ODE network is three hidden layers of size 64, with CELU activation in between layers. We trained for 20 local and 10 global epochs with sample size of 400 per time. We used the density loss with weight $\lambda_d = 20$.

**AML** The AML dataset is publicly available[5]. We first embed the data in 50 dimensions PCA space. We use an encoder of size $50, 8, 8$ with ReLU activations in between layers, trained for 1000 iterations with the RBF kernel (scale 0.5). For the trajectories we used an ODE network with three hidden layers of size $16, 32, 16$, with LeakyReLU in between layers, trained without noise for 50 local epochs with $\lambda_d = 20$.

## C  Additional Results

**Energy loss** In Fig. 6, we present the trajectories using the same parametrization as detailed in Sec. B with the energy loss ($\lambda_e = 1$). This loss requires much more memory since for each function evaluation we need to save the norm of the derivative. It should encourage smooth trajectory, but the accuracy of the trajectories is very sensitive to this loss. On Dyngen (left), it learns a straight line, and thus cannot bifurcate. On the petal, it initially learns a straight trajectory (middle, one epoch), and as training continues, it collapses in the center (right, ten epochs). Unless otherwise stated, we did not add it to our model since our trajectories were smooth. However, with careful parametrization, one could learn more accurate trajectories trajectories (see Fig. 7).

**Ablation and hold-out** To evaluate the prediction accuracy, we train by holding-out one timepoint, and we compute the distance between the held-out sample (ground truth) and the prediction. We use the $W_1$, the MMD with a Gaussian kernel, and the MMD with identity map ($L^2$ norm between the sample means). For the Petal and Dyngen datasets, we compute the Gaussian MMD by taking the average for the scales $0.1$ and $0.5$. For the EB data, we set the scale to $1500$.

In Fig. 8, we present an ablation study for the density loss and the geodesic embedding. The parametrization was found for the model with density loss and embedding, both seem important to learn accurate trajectories. In Fig. 9, we present the trajectories when we trained by holding one

---

[4] https://data.mendeley.com/datasets/v6n743h5ng/1
[5] https://www.ncbi.nlm.nih.gov/geo/query/acc.cgi?acc=GSE161676

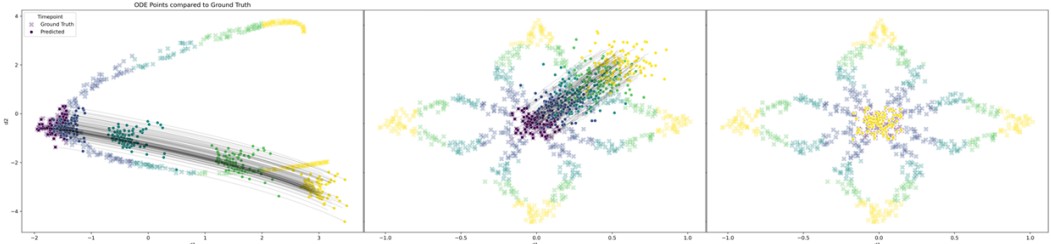

Figure 6: Trajectories with energy loss, and $\lambda_e = 1$.

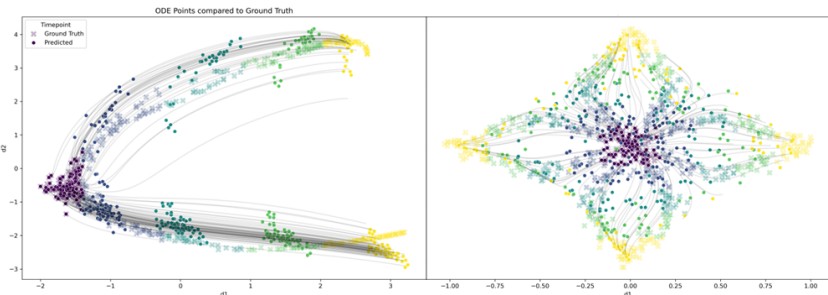

Figure 7: Trajectories with energy loss, and $\lambda_e = 0.01$.

timepoint out both with the embedding and density loss. In Fig. 10, we reproduce the same ablation on Dyngen data. Here, the density loss is important for the trajectories to stay on the manifold, and the embedding appears to help with the bifurcation. To complement this figure, in Tab. 6, we present the average of the KNN distance computed on the top 5%. On Dyngen, the trajectories with hold-out timepoints are presented in Fig. 11. Holding out the timepoint $t = 3$ seems to be the hardest to learn. In Tab. 3, we show the $W_1$ and MMD between the predicted values and ground truth, for MIOFlow, TrajectoryNet, and DSB. For the $W_1$, MIOFlow is always more accurate. In Tab 4, we compare the prediction accuracy on the EB data, depending on using the GAE or not. For the GAE, we consider the choice of two kernels when computing the distance $G$; the Gaussian kernel for various bandwidth parameters, and the $\alpha$-decay kernel for different $K$-nearest neighbors. Using the GAE improves the interpolation accuracy for all metrics. We embed the data in 200 PCA, and train the autoencoder with layers $200, 100, 100, 50$ for 1000 epochs. We trained the Neural ODE for 80 epochs with $\lambda_d = 20$, $\sigma_t = 0.2$, linear layers of size $16, 32, 16, 50$ with CELU activation in between layers. We train by holding out timepoint $t$ for the GAE and the neural ODE. For the MMD with the Gaussian kernel, we set the scale to $1500$.

Table 3: Leave-one-out $(t)$ $W_1$ and MMD with (G)aussian or (M)ean kernel between the predicted and ground truth point on the Petal and Dyngen datasets of MIOFlow, TrajectoryNet, and DSB. Lower is better.

|  |  | $t = 3$ | | | $t = 4$ | | | $t = 5$ | | |
|  |  | $W_1$ | MMD(G) | MMD(M) | $W_1$ | MMD(G) | MMD(M) | $W_1$ | MMD(G) | MMD(M) |
|---|---|---|---|---|---|---|---|---|---|---|
| Petal | MIOFlow(ours) | **0.170** | **0.051** | 0.002 | **0.200** | **0.052** | 0.015 | **0.218** | **0.055** | 0.009 |
|  | TrajectoryNet | 0.379 | 0.568 | 0.001 | 0.347 | 0.341 | 0.003 | 0.264 | 0.096 | 0.004 |
|  | DSB | 0.310 | 0.386 | <**0.001** | 0.260 | 0.087 | 0.029 | 0.441 | 0.159 | 0.081 |
|  | Baseline | 0.231 | 0.176 | 0.001 | 0.241 | 0.133 | **0.001** | 0.250 | 0.128 | **0.001** |
| Dyngen | MIOFlow(ours) | **0.509** | **0.181** | **0.022** | **1.787** | **0.324** | **3.071** | **1.450** | **0.491** | **1.719** |
|  | TrajectoryNet | 1.797 | 1.125 | 1.588 | 2.953 | 1.450 | 5.790 | 2.185 | 1.037 | 1.913 |
|  | DSB | 0.767 | 0.260 | 0.070 | 2.699 | 0.583 | 6.521 | 2.823 | 1.083 | 5.546 |
|  | Baseline | 1.828 | 0.958 | 2.435 | 2.198 | 1.131 | 3.368 | 2.221 | 1.133 | 3.139 |

**Nearest neighbor distance** To assist in evaluating how well the generated points land on the known manifold (ground truth points), we employ 1-NN distance from the generated points to the ground

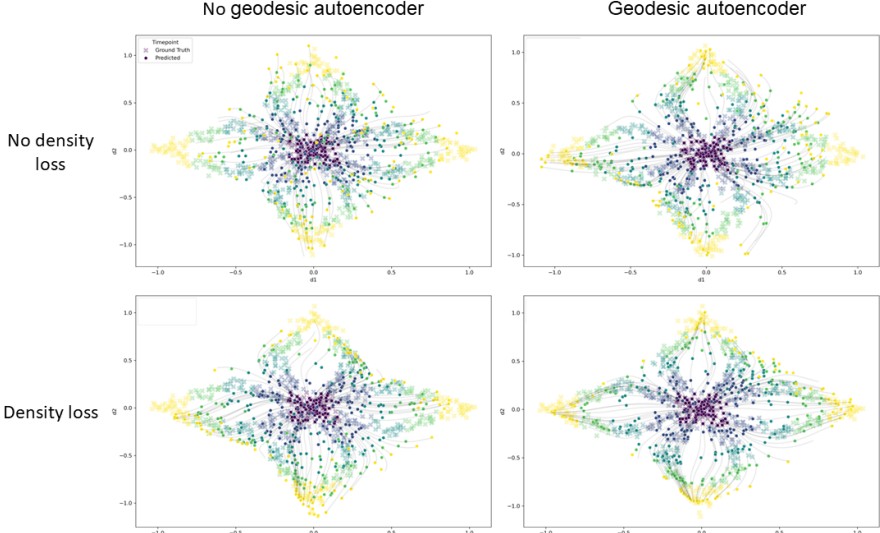

Figure 8: Petal dataset with and without the density loss (rows) or the geodesic autoencoder (columns).

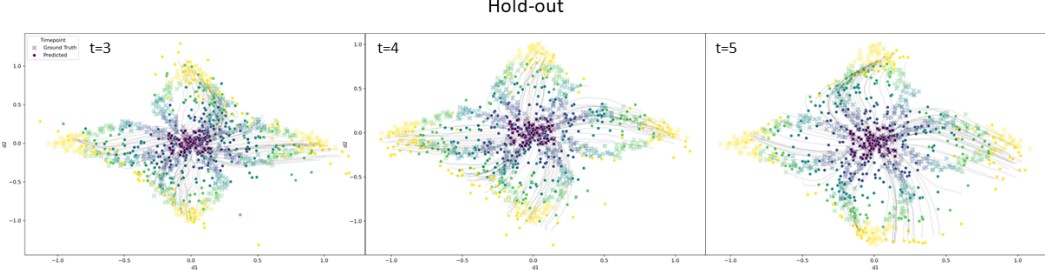

Figure 9: Trajectories for different hold-out time $t$ on the petal dataset.

truth points. Table 5 shows the 1-NN distance for several methods and datasets. Of note is the *aggregate* column. Where aggregate is specified as *mean*, we compute the mean 1-NN distance from the predicted point to the 1-NN at the same time index. Additionally, to help reduce sensitivity due to outliers, we computed these distances and report only the mean of the worst (highest 1-NN distance) quartile, where aggregate is specified as *quartile*.

**Neural SDE**   Additionally, we employed a Neural SDE from the PyTorch implementation of differentiable SDE Solvers [56, 55], and compared its results to MIOFlow (Fig. 12 & 13). We trained with a fixed diffusion function, but due to extensive training time (approximately two hours for 20 epochs for each dataset) we were unable to perform extensive hyperparameter optimization.

**DSB on petal**   In Fig. 14, we show prediction of the DSB algorithm for different ranges of noise scale $\gamma$. With too little noise, the prediction appear to collapse on a subset of the data, while the larger scales are inaccurate for this dataset.

## D  Software Versions

In Tab. 7, we present the GPU used during this work, and the following software version were used:     cudatoolkit==11.1.1, graphtools==1.5.2, numpy==1.19.5, phate==1.0.7 pytorch==1.10.2, pot==0.7.0, scikit-learn==0.24.2, scipy==1.5.3, scprep==1.1.0, torchdiffeq==0.2.2, torchsde==0.2.5

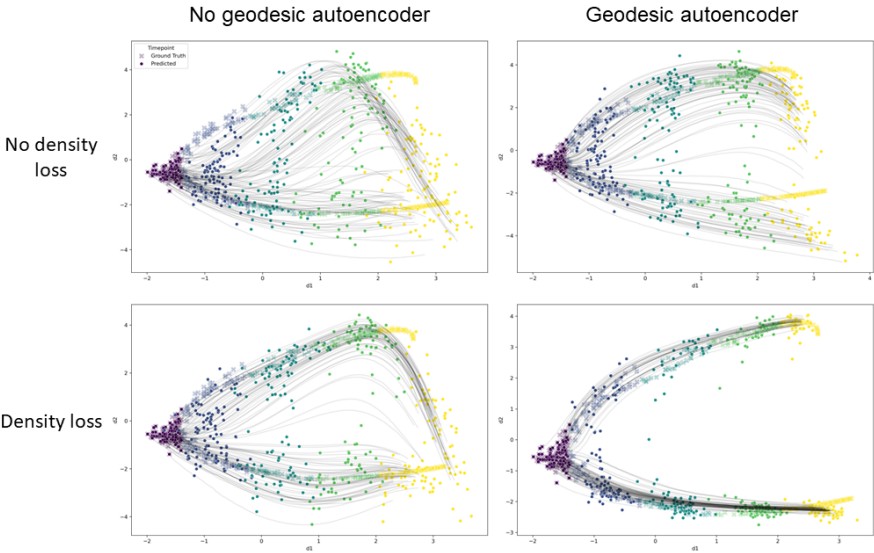

Figure 10: Dyngen dataset with and without the density loss (rows) or the geodesic autoencoder (columns).

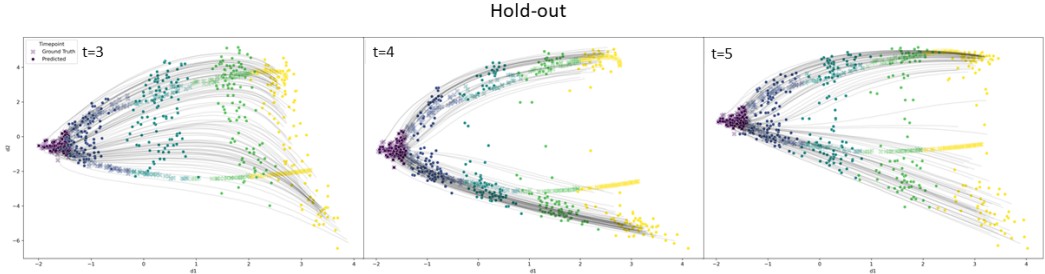

Figure 11: Trajectories for different hold-out time $t$ on the Dyngen dataset

Table 4: Leave-one-out $(t)$ $W_1$ and MMD with (G)aussian and (M)ean kernel between the predicted and ground truth point on the EB datasets for MIOFlow with the Gaussian, $\alpha$-decay kernel, or without GAE. Lower is better.

| | | | $t = 2$ | | | $t = 3$ | |
|---|---|---|---|---|---|---|---|
| GAE | | $W_1$ | MMD(G) | MMD(M) | $W_1$ | MMD(G) | MMD(M) |
| | $knn = 10$ | 25.774 | 0.072 | 119.386 | **32.129** | 0.135 | 206.144 |
| $\alpha$-decay | $knn = 20$ | **25.697** | 0.064 | 104.579 | 32.729 | 0.143 | 228.890 |
| | $knn = 30$ | 25.744 | 0.061 | **99.747** | 32.227 | 0.135 | 213.465 |
| | $\epsilon = 0.05$ | 26.873 | 0.064 | 115.905 | 32.844 | 0.132 | 213.900 |
| Gaussian | $\epsilon = 0.1$ | 26.107 | 0.064 | 109.653 | 32.776 | 0.123 | **203.617** |
| | $\epsilon = 0.5$ | 26.318 | **0.059** | 101.969 | 32.963 | 0.138 | 223.679 |
| No GAE | | 29.243 | 0.063 | 126.608 | 35.709 | 0.119 | 246.609 |
| Baseline | | 33.415 | 0.103 | 227.279 | 35.319 | **0.095** | 213.492 |

Table 5: Average 1-NN distance of predicted points from a given method to ground truth points at the same time label across two datasets. Rows designated *mean* are the average 1-NN distance, while those designated *quartile* are the average 1-NN distance for the worst quartile. Lower is better.

|  | MIOFlow(ours) | TrajectoryNet | DSB | aggregate |
|---|---|---|---|---|
| Petal | **0.033** | 1.823 | 0.286 | mean |
| Dyngen | **0.586** | 2.562 | 1.638 | mean |
| Petal | **0.096** | 2.769 | 0.618 | quartile |
| Dyngen | **1.163** | 4.277 | 3.474 | quartile |

Table 6: On the Dyngen dataset, average distance of the 10-NN for predicted points between ground truth at the same time or any. The average is computed on the highest $5\%$ observations. Lower is better.

| GAE |  | time | any |
|---|---|---|---|
| $\alpha$-decay | knn$= 5$ | 19.922 | 19.249 |
|  | knn$= 10$ | 19.987 | 19.097 |
|  | knn$= 15$ | 18.579 | 16.905 |
| Gaussian | $\epsilon = 0.05$ | 16.245 | 14.972 |
|  | $\epsilon = 0.1$ | 16.245 | 15.491 |
|  | $\epsilon = 0.5$ | **15.427** | **14.101** |
| No GAE | Euclidean | 21.589 | 20.486 |

a) Neural SDE                           b) MIOFlow (ours)

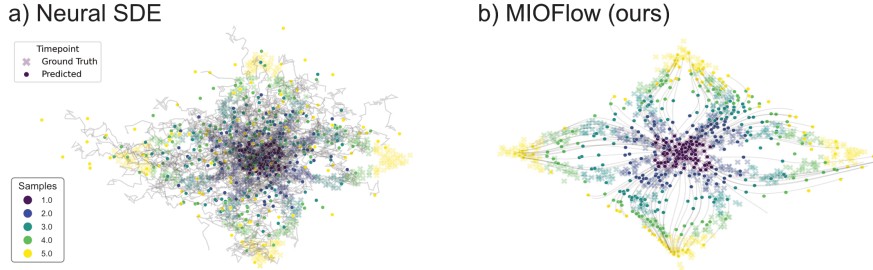

Figure 12: Comparison of Neural SDE (a) to MIOFlow (b) on the petal dataset.

a) Neural SDE                           b) MIOFlow (ours)

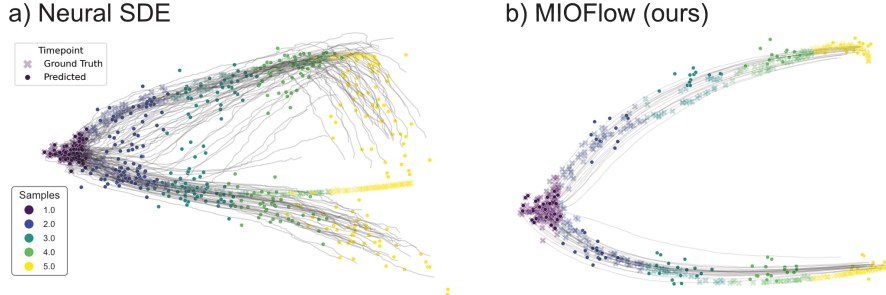

Figure 13: Comparison of Neural SDE (a) to MIOFlow (b) on the dyngen dataset.

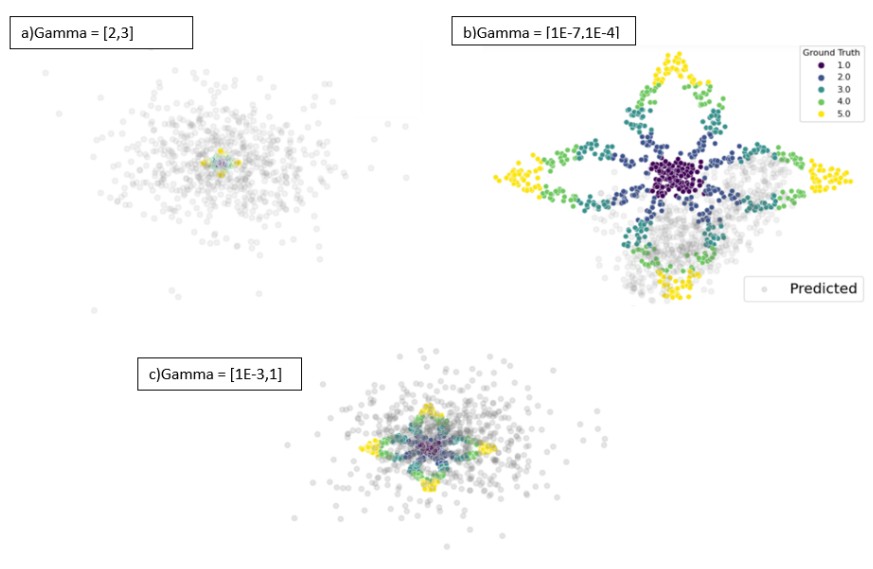

Figure 14: Comparison of DSB for different values of the noise scale $\gamma$.

Table 7: GPU specifications of High Performance Cluster. Results were generated over a swath of GPU generations and models.

| Count | CPU Type | CPUs/Node | Memory/Node (GiB) | GPU Type | GPUs/Node | vRAM/GPU (GB) | Node Features |
|---|---|---|---|---|---|---|---|
| 1 | 6240 | 36 | 370 | a100 | 4 | 40 | cascadelake, avx2, avx512, 6240, doubleprecision, common, bigtmp, a100 |
| 6 | 5222 | 8 | 181 | rtx5000 | 4 | 16 | cascadelake, avx2, avx512, 5222, doubleprecision, common, bigtmp, rtx5000 |
| 4 | 5222 | 8 | 181 | rtx3090 | 4 | 24 | cascadelake, avx2, avx512, 5222, doubleprecision, common, bigtmp, rtx3090 |
| 8 | E5-2637_v4 | 8 | 119 | gtx1080ti | 4 | 11 | broadwell, avx2, E5-2637_v4, singleprecision, common, gtx1080ti |
| 2 | E5-2660_v3 | 20 | 119 | k80 | 4 | 12 | haswell, avx2, E5-2660_v3, doubleprecision, common, k80 |

## Supplement References

[54] Marco Cuturi. Sinkhorn Distances: Lightspeed Computation of Optimal Transport. In *Advances in Neural Information Processing Systems*, volume 26. Curran Associates, Inc., 2013.

[55] Patrick Kidger, James Foster, Xuechen Li, Harald Oberhauser, and Terry Lyons. Neural SDEs as Infinite-Dimensional GANs. *International Conference on Machine Learning*, 2021.

[56] Xuechen Li, Ting-Kam Leonard Wong, Ricky T. Q. Chen, and David Duvenaud. Scalable gradients for stochastic differential equations. *International Conference on Artificial Intelligence and Statistics*, 2020.

[57] Ilya Loshchilov and Frank Hutter. Decoupled weight decay regularization. In *International Conference on Learning Representations*, 2018.

[58] Richard Sinkhorn. A relationship between arbitrary positive matrices and doubly stochastic matrices. *The annals of mathematical statistics*, 35(2):876–879, 1964.

[59] Feng-Yu Wang. Sharp explicit lower bounds of heat kernels. *The Annals of Probability*, 25(4):1995–2006, 1997.