# OpenReview forum: "Manifold Interpolating Optimal-Transport Flows for Trajectory Inference"
_NeurIPS.cc/2022/Conference — NeurIPS 2022 Accept_

### Official Review · Reviewer_JuWZ · 2022-07-09

**Rating:** 7
**Confidence:** 2
**Soundness:** 3 good
**Presentation:** 3 good
**Contribution:** 3 good

**Summary:**

The authors propose a new architecture, called Manifold Interpolating Optimal-Transport Flow (MIOFlow), to learn the dynamics of snapshot data like single-cell data. It consists of two major parts: 1. An autoencoder that embeds the data into a latent space, conserving a geodesic distance. 2. The MIOFlow that is based on neural ordinary differential equations (nODE), which is trained to model the stochastic trajectories of snapshot data. To this end the embedding of the autoencoder is combined with noise as the input to the dynamics function of the nODE.
They evaluate their model on artificial data as well as cell real word single-cell data, showing the advantages of their method.


**Questions:**

Minor comment: Some sentences have too many/wrong words. Examples include:
- line 18
- Caption of Figure one

**Limitations:**

Yes

**Strengths And Weaknesses:**

Strengths:
- To my knowledge, the MIOFlow architecture is a novel idea.
- The manuscript is clearly and well written.
- They compare MIOFlow with other models and show superior performance on several datasets including single-cell data.

Weaknesses:
- Most results are qualitative rather than quantitative

---

> ### Author Response · Authors · 2022-08-02
> **Response to Reviewer JuWZ**
>
> We thank the reviewer for their positive review and helpful feedback. We respond to some questions raised by the reviewer.
>
> > Most results are qualitative rather than quantitative
>
> We have more quantitative results in the supplementary material see Tab 2-5. We added two experiments with quantitative evaluation. On the Dyngen dataset, we evalute how close the trajectories are to the manifold, whether we train with or without the geodesic embedding(GAE) for different kernels and parameters. We will add the same experiment for the Petal dataset. We also added a hold-out experiment on the EB dataset for different choice of kernels.
>
>
> Thank you for your minor corrections. We have fixed these in the updated draft.

---

### Official Review · Reviewer_j3Kk · 2022-07-10

**Rating:** 5
**Confidence:** 1
**Soundness:** 3 good
**Presentation:** 3 good
**Contribution:** 3 good

**Summary:**

The paper proposes a computationally efficient dynamic optimal transport framework to track samples overtime.




**Questions:**

- in Theorem 1: what is $d^{2\alpha}_{\mathcal{M}}$ ?
- what is the meaning of the sentence preceding Theorem 1 mentioning "Thm.2" ?



**Limitations:**

The limitations are well addressed. The proposed method works best with uniform sampling in the time dimension and also consider balanced optimal transport (equal mass of density functions whatever time t).

**Strengths And Weaknesses:**

Overall the paper is well written.

Some elements are unclear however
e.g.
- Fig 1 caption "Geodesic Autoencoder learns the latent space of representation of data is learnt such that is preserves a diffusion geodesic distance."  (?)

- the notation "d" is used for dimension and also for distances.

- some element of section 3 are well known and may not need to be written (e.g. wasserstein distance equation),

- section 3.3 is unclear.

- the connection between section 3  and section 4 is not clear.

---

> ### Author Response · Authors · 2022-08-02
> **## Response to Reviewer j3Kk**
>
> We thank the reviewer for their positive review particularly their remark that the paper was overall well written. We have fixed the minor comments in the draft and we respond to questions raised by the reviewer here.
>
> >Fig 1 caption "Geodesic Autoencoder learns the latent space of representation of data is learnt such that is preserves a diffusion geodesic distance." (?)
>
> This caption has been rewritten to say "... The Geodesic Autoencoder learns a latent space that preserves the diffusion geodesic distance ..."
>
> >the notation "d" is used for dimension and also for distances.
>
> $d$ is now used only for distance, we use $k$ for dimension.
>
> >some element of section 3 are well known and may not need to be written (e.g. wasserstein distance equation),
>
> Since this work combines many fields, we have aired on the side of providing too much background rather than too little.
>
> >section 3.3 is unclear
>
> Section 3.3 has been rewritten for clarity. Note that it now corresponds to section 3.4.
>
> >the connection between section 3 and section 4 is not clear.
>
> In section 3, we review some theoretical results that are needed to define our method in section 4. To strengthen the link between the two sections, we added a subsection on manifold learning and more details on the motivation of using a diffusion geodesic for optimal transport.
>
> > in Theorem 1: what is $d_\mathcal{M}^{2 \alpha}$?
>
> $d_\mathcal{M}^{2 \alpha}$ represents the Riemannian manifold geodesic distance powered to $2 \alpha$. This is necessary as while Theorem 1 shows $D_\alpha$ is equivalent to $d_\mathcal{M}^{2 \alpha}$ for $\alpha \in (0, 1/2)$ including for $\alpha$ arbitrarily close to $1/2$, the equivalence breaks down at $\alpha = 1/2$.
>
> > what is the meaning of the sentence preceding Theorem 1 mentioning "Thm.2" ?
>
> This theorem is proved in Leeb and Coifman 2016. We have added a clarification to the theorem statement. We have included it as it is useful both to understand and prove our subsequent results in Theorem 2 and Corollary 1.

---

### Official Review · Reviewer_JVYo · 2022-07-12

**Rating:** 7
**Confidence:** 3
**Soundness:** 3 good
**Presentation:** 3 good
**Contribution:** 3 good

**Summary:**

The authors propose a geodesic auto encoder to map points on a manifold to latent space whilst preserving geodesic distance on the data manifold in Euclidean distance in the latent space.

The geodesic auto encoder is then used to simulate trajectories in latent space, using diffusions and Neural ODEs. One may decode back to the data space. The trajectories are trained through an OT loss to samples.

The authors show a number of theoretical results to justify the preservation of geodesic loss in latent space, and how Wasserstein in latent space with L2 ground cost is equivalent to Wasserstein on the decoded space with geodesic ground cost.

Experiments are carried out on biological data to justify the approach, and method seem to work well and results are convincing.

**Questions:**

- In Table 1, what does runtime refer to? Is it train time or runtime at deployment?
- DSB interpolates via reverse diffusions and approximates the OT cost between initial and terminal distributions (no intermediary distributions) in Euclidean space where the ground cost is determined by the initial forward diffusion (reference diffusion). Which reference diffusion is used here?
- It is stated in the appendices that DSB is used with "default" parameters, what are these default parameters? What network was used? How many diffusion steps?
- The diagrams are not clear to me. Are the colours points labelled "samples" samples from the training data or the model? How come Figure 2 a and c have grey lines but Figure 2b does not? The legend states ground truth is labelled by a cross and the predicted by a filled in dot, then how come the ground truth varies so much by the method used to interpolate?
- What layer widths were used for the Petal and Dyngen experiments and what was the latent dimension?

**Limitations:**

The authors acknowledge training ODE may be challenging in some settings and that they use <10 time steps.

**Strengths And Weaknesses:**

Strengths
- Training a geodesic auto encoder appears novel, has a lot of applications, and allows standard machine learning tools in Euclidean space to be applied to data living on a manifold. The method presented is simple and easy to understand.
- The geodesic auto encoder + trajectory with Neural ODE/ SDE approach presented seems quite flexible and able to be modified to intermediate distributions, unlike the Schordinger bridge, which has practical importance
- The method seem to work well and results are convincing
- Interesting theoretical results connecting intermediate trajectories to OT

Weaknesses
- It is not clear how well the decoder generates data close to the manifold in question. Comparison to manifolds with analytic form such as a sphere/ hypersphere may be helpful to compare to a ground truth.
- [1] appears highly relevant and should perhaps be used as a baseline, especially given the similarities using W2 loss to map cross sectional samples of a time evolving population
- The diagrams are not clear to me. The coloured points are labelled "samples", are these samples from the training data or the model? How come Figure 2a and Figure2c have grey lines but Figure 2b does not? The legend states ground truth is labelled by a cross and the predicted by a filled in dot, then how come the ground truth varies so much by method used to interpolate?
- The link to code does not work: https://anonymous.4open.science/r/bananarama-FADD/README.md
- It is not clear why the geodesic auto encoder does not work well for a large N, as mentioned in line 191
- A written algorithm would be helpful to understand the training procedure

- Experiments
    - It appears prediction at time 3,4 in the Appendices Table 2 for leave one out prediction are quite different to the results in Table 1 in the main text. It may give a clearer, more balanced view to include all the results. Also why was W1 chosen over W2?
    - Number of time steps is relatively small


Minor:
- "sinkhorn" is a name and should be capitalised in line 79
- line 310, should be "compared"

[1] Proximal Optimal Transport Modeling of Population Dynamics, Bunne, 2021

---

> ### Author Response · Authors · 2022-08-02
> **Response to Reviewer JVYo Part 1**
>
> We thank the reviewer for their positive review particularly their remark that the paper was overall well written. We have fixed the minor comments in the draft and we respond to questions raised by the reviewer here.
>
> >It is not clear how well the decoder generates data close to the manifold in question. Comparison to manifolds with analytic form such as a sphere/ hypersphere may be helpful to compare to a ground truth.
>
> Thank you for this helpful suggestion! First we would like to point out that we don't always have to use the reconstruction loss. For low dimensional data, we can use the encoder only to leverage its distance property, than we use the encoded data to compute the Wasserstein loss. Otherwise, when we use the Geodesic AutoEncoder (GAE), we add the decoder and the reconstruction loss (mean squared error). During training we also add noise to the inputs which potentially bring them off the manifold. The decoder learns to denoise the inputs, and bringing them to their original position. The reconstruction penalty decreases even though we also train to preserves the diffusion geodesic distance. For example, on the Dyngen dataset, the reconstruction loss is initially at `1.821` and goes to `0.007` after a thousand iterations.
>
> Similar ideas were introduced in [2,3] where the authors penalize an autoencoder to match a certain manifold learning embedding, either PHATE or Diffusion map. Here, we regularize the embedding with a novel distance which is also based on the diffusion probabilities.
>
>
> >[1] appears highly relevant and should perhaps be used as a baseline, especially given the similarities using W2 loss to map cross sectional samples of a time evolving population
>
> Thank you for mentioning this interesting work. JKONets [1] describes a new formulation to solve for the energy function assuming the population data evolves according to a JKO flow. We beleive this is a promising direction especially as the energy function has an additional benefit in terms of interpretability. However, it is unclear how to bias the JKONet to perform transport along a manifold as flows are not directly represented. something like our density loss (encouraging flows close to data) is not computationally tractable as there is no explicit flow to integrate over.
>
> JKONet provides an alternative way to solve for the Euclidean OT transport flows over time and could be used instead of a continuous normalizing flow on top of our Geodesic autoencoder, we leave exploration of this direction to future work.
>
> Due to code complications we are not able to provide a direct comparison at this time, but will continue to try over the next week. We expect JKONets will work well in interpolating Euclidean flows, but may fail to follow the data manifold as it has no reason to do so.
>
>
> >The diagrams are not clear to me.
>
> The caption of Fig.1 wasn’t clear and we rewrote it, we also modified the figure to accentuate the link between the geodesic autoencoder and the Neural ODE. We modified the figures 2 and 3 so that it is easier to identify the ground truth, the predicted values and the trajectories. Now the ground truth points are designated by translucent `X` in the background, whereas predicted points are opaque circles, whilst trajectories are solid lines.
>
> >The link to the code does not work:
>
> Thank you for pointing this out and for trying to check out the code. Sorry about that! We transferred ownership of the underlying repository, which apparently breaks the `anonymous.4open.science` link. The old link cannot be fixed, the updated link is:
> https://anonymous.4open.science/r/bananarama-FADD/
>
> > It is not clear why the geodesic auto encoder does not work well for a large N, as mentioned in line 191
>
> Note that $G$ is our realization of $D_\alpha$, and in particular $G$ hides a fixed maximum scale parameter $K$ (as discussed in the supplement), and is **not** the geodesic autoencoder. If we take large enough $K$ then $G$ will capture meaningful distances. However, with a fixed $K$, as $n \to \infty$ our approximation of $D_\alpha$ worsens. The problem occurs when $G$ is computed on the **entire** dataset. By encoding the distances $G$ into an autoencoder we can alleviate the computational problem since we use batches of the data to compute $G$ and train the autoencoder.
>
> This sentence is overly confusing and not central to our argument, and so we have removed it in the updated draft and replaced it with "Computing $G_\alpha$ on the entire dataset is inefficient due to the powering of the diffusion matrix. We circumvent this difficulty with the encoder, since we train on subsamples."
>
> > A written algorithm would be helpful to understand the training procedure
>
> Thank you for the suggestion, we will add an algorithm either in the main text or in the supplement.

---

> > ### Author Response · Authors · 2022-08-02
> > **Response to Reviewer JVYo Part 2**
> >
> >
> > >It appears prediction at time 3,4 in the Appendices Table 2 for leave one out prediction are quite different to the results in Table 1 in the main text. It may give a clearer, more balanced view to include all the results. Also why was W1 chosen over W2?
> >
> > We choose $W_1$ over $W_2$ because we wanted a distance that was not used during training (we train with $W_2$). We did not include all the results due to the page constraint, but given an extra page, we could add the other hold-out timepoints.
> >
> > > Number of time steps is relatively small
> >
> > We agree that the number of time steps is relatively small. In the single-cell setting this reflects realistic measurement types as each sample over time adds additional cost and complexity. To get a sense of scale, the dataset with the largest number of time steps to our knowledge is 18 [Schiebinger et al. 2018], which was specifically collected with Euclidean OT interpolation in mind. Much more common are datasets in the 4-10 time step range. We therefore aim to test at this range of time steps. For instance, 100 or 1000x more time steps may make manifold penalties irrelevant on Riemannian manifolds as the locally Euclidean assumption may mean Euclidean OT does well enough. For the time being however, current high quality datasets make the manifold assumptions necessary in OT-based interpolation.
> >
> > > In Table 1, what does runtime refer to? Is it train time or runtime at deployment?
> >
> > It refers to the training time, for our method it includes both the training of the antoencoder and the training of the Neural ODE. We changed the caption of Table 1.
> >
> > > DSB interpolates via reverse diffusions and approximates the OT cost between initial and terminal distributions (no intermediary distributions) in Euclidean space where the ground cost is determined by the initial forward diffusion (reference diffusion). Which reference diffusion is used here?
> >
> > Our reference diffusion is a multivariate gaussian with the same dimension as the drift term. We learn the scale parameter $\sigma_t$ for each discretization time $t$. We also implemented our model with a SDE solver from the package `torchsde`, allowing to consider more complex diffusion terms and different reference measures.
> >
> > > It is stated in the appendices that DSB is used with "default" parameters, what are these default parameters? What network was used? How many diffusion steps?
> >
> > A score network with 2 MLP encoders (one for time, one for position) and a subsequent MLP decoder network that takes that concatenated time and position embedding as input and outputs a new embedding was used. 20 Diffusion time steps and 5000 iterations were the default settings under which the experiment was conducted.
> >
> > >What layer widths were used for the Petal and Dyngen experiments and what was the latent dimension?
> >
> > For both datasets, we used two linear layers of size 8 and 32. We added these detail in the experiment section of the supplementary material.
> >
> > [2] Gal Mishne, Uri Shaham, Alexander Cloninger, and Israel Cohen. Diffusion nets. Applied and Computational Harmonic Analysis, 2019
> >
> > [3] Andrés F Duque, Sacha Morin, Guy Wolf, and Kevin Moon. Extendable and invertible manifold learning with geometry regularized autoencoders. IEEE International Conference on Big Data. 2020.

---

> > > ### Comment · Reviewer_JVYo · 2022-08-08
> > > **Summary**
> > >
> > > Overall, this seems like a reasonable contribution.
> > > - Methodologically the main contribution is primarily the geodesic autoencoder, though perhaps needs more empirical validation that it is actually generating points on a manifold.
> > > - The training regimes are quite simple, consisting of multiple loss terms for geodesic encoding + neural sde. Though a written algorithm would improve clarity.
> > > - On reflection, a better job could have been done comparing with baselines such as:
> > >
> > > [1] Proximal Optimal Transport Modeling of Population Dynamics, Bunne, 2021 \
> > > [2] Optimizing Functionals on the Space of Probabilities with Input Convex Neural Networks, Melis 2021 \
> > > [3] Neural Lagrangian Schrödinger Bridge,   11 Apr 2022 (perhaps too recent to compare)
> > >
> > > Minor:
> > > - **A score network with 2 MLP encoders (one for time, one for position) and a subsequent MLP decoder network that takes that concatenated time and position embedding as input and outputs a new embedding was used. 20 Diffusion time steps and 5000 iterations were the default settings under which the experiment was conducted.**
> > >
> > > How many layers in the MLP and what width?  These details are important and should be included.
> > >
> > > - **Our reference diffusion is a multivariate gaussian with the same dimension as the drift term.**
> > >
> > > A multivariate gaussian is not a diffusion. Do you mean Brownian motion? Do you mean OU process? What is the initial forward drift function?
> > >
> > > - **Now the ground truth points are designated by translucent X in the background, whereas predicted points are opaque circles, whilst trajectories are solid lines.**
> > >
> > > Figure 2 seems incorrect, DSB is able to generate complex images. It seems odd that the final trajectories are so far from the ground truth at the terminal values. Following the authors' previous comments, this seems like an implementation error. **a SDE solver from the package torchsde, allowing to consider more complex diffusion terms and different reference measures** it is not clear how torchsde will work with DSB.
> > >
> > > For these small scale problems it may also be worth using the Gaussian Process Schrodinger Bridge implementation rather than neural networks, which are harder to train see e.g. [1]
> > > Solving Schrödinger Bridges via Maximum Likelihood, Vargas 2021
> > > https://arxiv.org/abs/2106.02081

---

> > > > ### Author Response · Authors · 2022-08-09
> > > > **Response to minor comments**
> > > >
> > > > Thank you for your insightful comments. We will add an algorithm description to improve clarity.
> > > >
> > > > **Minor**
> > > >
> > > > >How many layers in the MLP and what width? These details are important and should be included.
> > > >
> > > > The MLPs for the input and time are two layers of size 16 and 32. They are then concatenated and used as inputs for an MLP with two layers of size 128 each, all separated by `LeakyReLU`. More details can be found in the *basic* model [here](https://github.com/JTT94/diffusion_schrodinger_bridge/blob/main/bridge/models/basic/basic.py).
> > > >
> > > > >A multivariate gaussian is not a diffusion. Do you mean Brownian motion? Do you mean OU process? What is the initial forward drift function?
> > > >
> > > > The reference is a Brownian motion, the initial drift is a random initialization of the neural network.
> > > >
> > > > >Figure 2 seems incorrect, DSB is able to generate complex images. It seems odd that the final trajectories are so far from the ground truth at the terminal values. Following the authors' previous comments, this seems like an implementation error.
> > > >
> > > > We agree that the DSB trajectories in Fig.2 look underfit, possibly due to too much noise. We use the author's implementation https://github.com/JTT94/diffusion_schrodinger_bridge, with the default parameters, it is possible that these parameters are not optimal for our datasets. Their parameters were tuned on datasets with different scales, we will run the experiments again with a range of new noise scale parameters.
> > > >
> > > > >it is not clear how torchsde will work with DSB.
> > > >
> > > > We mentioned `torchsde` for our method not for the DSB. Our implementation supports `torchsde`.

---

> > ### Comment · Reviewer_JVYo · 2022-08-03
> > **code link**
> >
> > **The old link cannot be fixed, the updated link is: https://anonymous.4open.science/r/bananarama-FADD/**
> >
> > Isn't this the same link? It also does not seem to work.

---

> > > ### Author Response · Authors · 2022-08-03
> > > **Correct link**
> > >
> > > Thank you for pointing this out again. The following link works [https://anonymous.4open.science/r/bananarama-FAD/README.md](https://anonymous.4open.science/r/bananarama-FAD/README.md). Currently, the tutorials are not accessible, but you do have access to some notebooks. Most importantly, you can access the functions and classes files on the left of the page. For example, the models are defined in `03_models.ipynb`, and the losses in `01_losses.ipynb`.

---

### Official Review · Reviewer_CfAn · 2022-07-21

**Rating:** 4
**Confidence:** 4
**Soundness:** 1 poor
**Presentation:** 1 poor
**Contribution:** 2 fair

**Summary:**

The authors consider a problem of modeling distribution evolution given samples for different discrete time moments.

The paper consists of several parts.

First, the authors describe some concepts on how to construct an autoencoder that preserves distances over a Riemannian manifold on which the data is located. They define the diffusion ground distance on the manifold. And then describe an approximation of the diffusion geodesic distance. Based on this approximation, we can estimate the diffusion ground distance and train an autoencoder such that the L2 distance in the latent space approximates these estimates.

Second, the authors introduce a definition of OT based on a time-evolving vector field and on a diffusion formulation (see (3) and (4)). In Th. 3 they repeat a theorem from [35] that OT between the measures mu and nu can be represented as an optimization w.r.t. the vector field of an integral term plus some regularization.

Third, the authors model the vector field with a neural ODE and train the parameters of a neural ODE so that between each consecutive step the neural ODE solves the OT problem, formulated based on the diffusion formulation.

Finally, the authors applied the proposed algorithm to model some trajectories of cells.

**Questions:**

- 18: “However, most have these works have focused on generative modeling” - bad wording

- TrajectoryNet, and other papers, mentioned in the introduction, are not the only approaches to model stochastic flows. For example, what about the approaches from papers
1) Large-Scale Wasserstein Gradient Flows
Petr Mokrov, Alexander Korotin, Lingxiao Li, Aude Genevay, Justin Solomon, Evgeny Burnaev
Neurips, 2021
2) Charlie Frogner and Tomaso Poggio. Approximate inference with Wasserstein gradient flows. AISTATS, 2020

- The authors use a geodesic autoencoder. Any particular motivation why for using such an autoencoder? Why not some standard autoencoder? Any comparison between using the geodesic autoencoder and the standard one? How does this choice influences the experimental results?

- 66: “an initial state [15, 38] in can be combined to infer the cell lineage…” - in can be? Looks like a misprint.

- 91: Here, d1 and d2 are distances. Earlier, d was used as a dimension. Moreover, again in 104 d(x,y) denotes a distance. I propose to use some consistent notations.

- Why consider a Riemannian manifold? What if the manifold is not a Riemannian one? What is the motivation? How does this choice influences the experimental results?

- The authors use the heat kernel and estimate the diffusion ground distance using the Gaussian kernel function. What if we use some other kernel function? Will the choice significantly influence the accuracy of the approximation and final results? How does this choice influences the experimental results?

- How does a value of alpha influence the experimental results?

- It is unclear to what extent the diffusion geodesic distance approximates anything accurately. Any comments on the accuracy of approximation in Th. 2? Also, the approximation here is stochastic. So the formulation should be either w.r.t. probability or a.s. Any comments about this?

- The geodesic distance is introduced in 139, but there is no exact definition.

- 153: mu is an actual distribution on M. Is it connected somehow with mu in (3)?

- How does the choice of beta in 153 influence the experimental results?

- Why did the authors use Neural ODE? Why not standard RNN? Any rationale behind the choice?

- Fig. 1: “With the combination of noise, a neural network learns to predict the derivative x′(t0), and an ODE Solver produces the predicted x(t1).” - x(t) is a stochastic process. Which derivative do the authors mean? Also, nothing was told about using the derivatives w.r.t the trajectories in the paper's main text.

- Fig. 1: “Geodesic Autoencoder learns the latent space of representation of data is learnt such that is preserves a diffusion geodesic distance. ” - is this crucial for the considered applications?

- 182: “With a slight abuse of notation, a similar distance between distributions (rather than points) is constructed in [37] where it was applied to approximate the Earth Mover’s Distance with a geodesic ground cost.” What is the novelty of this approach?

- 186: instead of = we should place \approx here.

- 189: “we use it to compute distances between predicted and ground truth observations.” Which ground truth observations?

- 191: “Empirically, we observed that G fails to capture meaningful distances for a large number
 of observations. ” - if it fails, then what is the point of using it?

- 202: “to understand how specific genes evolved over time, by decoding in the gene space.” - I do not understand the paper's main topic then. Do the authors propose a new method? Or is this about the modeling of genes? or what?

- (6): W_d -> W_2

- Th. 3. Which D should we consider? Is any D OK?

- 224: too many indexes i are used in the same formula. I guess a summation should be done w.r.t. to some other index.

- (7): How do the authors differentiate L_m w.r.t. parameters of the Neural ODE?

- 238: T -> $T$

- 260: “trajectories on the manifold”. Why are they located on any manifold? How should the reader analyze Fig. 2 and conclude that the results for the proposed method are better? What is the ground truth? Any accuracy characteristics to compare methods?

- Table 1: Are the numbers in the table big or small? We should compare this with some simple baseline approaches. Possibly all methods provide some bad results?

- 293: “[39], Other …” -> “[39]. Other …”

- Appendix, 519: “Replacing the continuity equation by the path dynamic, we have…” Why can we make such a replacement? Any justification that their proof is valid for any dissimilarity?

- Appendix, 520-521: Here, the authors wrote that the autoencoder approximates the geodesic distance for sample points. First, this is not completely true, as in Th. 2 we also have a parameter alpha. Also, if these quantities are equal for sample points, it does not follow that for any pair of points from some distribution, the autoencoder will have good generalization ability so that inequalities in 522 are valid.

**Limitations:**

The potential negative societal impact of the work is not presented in the paper, although the authors did not explicitly state this.

Some limitations of the paper are addressed.

**Strengths And Weaknesses:**

Strengths
- The topic of the paper is about modeling stochastic dynamics, which is important

Weaknesses
- The text of the paper is of low quality and clarity, and contains errors and misprints, see comments below.
- The paper is a combination of several moving pieces, and it is unclear why exactly such choices were made.

---

> ### Author Response · Authors · 2022-08-02
> **Response to Reviewer CfAn part 1**
>
> We would like to thank the reviewer for the careful examination of our manuscript. We will first summarize the main misunderstandings and then address each point individually.
>
> **Motivation for considering manifold structure in data**
> The reviewer asked why the geodesic autoencoder is necessary. We would like to clarify that the various components of our manuscript are motivated by the key problem we are tackling: *implementing efficient flows on data with manifold structure*. This is a problem that naturally arises in cellular and other naturalistic systems, where there are static snapshot measurements from which dynamics must be inferred, moreover it has been established repeatedly that such data follows the manifold assumption [6,7,8,9].
>
> Since we are focused on data with manifold structure, we operate in the intrinsic manifold dimensions instead of in the ambient space, and the Geodesic autoencoder is necessary for this. Next, in order to learn flow dynamics from static snapshot data, we utilize a neural ODE. Finally, since natural flow dynamics are rarely circuitous (or wasteful of energy), we penalize the neural ODE into performing efficient transport.
>
> When viewed from our original motivation our choices are natural.  We added motivation for solving the problem of optimal transport on a manifold in section 1 and 3.4. Though not explicitly formulated as such other papers have also attempted to use the heat kernel to perform optimal transport, which naturally follows data manifolds, see for example [1,2,3].
>
> **Use of the Riemannian Manifold Assumption**
> The reviewer asked about the Riemannian manifold assumption. We would like to clarify that this is a technical assumption necessary for the asymptotic proofs and does not affect the experimental results.
>
> **Point by point responses**
> >The text of the paper is of low quality and clarity, and contains errors and misprints, see comments below.
>
> Thank you for taking the time to point out misprints, and we apologize for the lack of clarity in certain sections of the manuscript. We fixed all the misprints pointed out in the review and we made the links between each section clearer.
>
> >The paper is a combination of several moving pieces, and it is unclear why exactly such choices were made.
>
> As addressed above, there are two major components, first the geodesic autoencoder, which helps to learn flows on manifold data, and second the neural ODE which learn the trajectories in the embedded space. Our choices were made to optimize our main task: interpolating flows on data with manifold structure.
>
> > The authors use a geodesic autoencoder. Any particular motivation why for using such an autoencoder? Why not some standard autoencoder? Any comparison between using the geodesic autoencoder and the standard one? How does this choice influences the experimental results?
>
> We proved that the geodesic autoencoder reflects geodesic distances under some conditions. This does not hold for a standard autoencoder, where the relationship between distances in the latent space and distances in the ambient space is unknown.
>
> Standard autoencoders are only constrained by the reconstruction penalty, and they can learn any compression that can be decoded based on the specifics of the architecture. Previous work [13,14] has shown that manifold-regularizing autoencoders results in a manifold embedding.
>
>
> > Here, d1 and d2 are distances. Earlier, d was used as a dimension. Moreover, again in 104 d(x,y) denotes a distance. I propose to use some consistent notations.
>
> We now use the notation $k$ for dimension.
> > Why consider a Riemannian manifold? What if the manifold is not a Riemannian one? What is the motivation? How does this choice influences the experimental results?
>
> Considering a closed Riemannian manifold is a technical condition, and does not affect the experimental results. This assumption is essential for proving convergence results of the discrete heat operator to the continuous one. If the manifold is not Riemannian then we cannot prove that the discrete distance $D_\alpha$ that we optimize for converges to the continuous manifold geodesic $d_\mathcal{M}$ (as on a general manifold $d_\mathcal{M}$ may not even exist).
>
> In practice, discrete diffusion-based distances are widely successful in applications outside of strictly Riemannian manifolds. $D_\alpha$ is still well defined, but is easiest to interpret under this technical condition.

---

> > ### Author Response · Authors · 2022-08-02
> > **Response to Reviewer CfAn part 2**
> >
> > > The authors use the heat kernel and estimate the diffusion ground distance using the Gaussian kernel function. What if we use some other kernel function? Will the choice significantly influence the accuracy of the approximation and final results? How does this choice influences the experimental results?
> >
> > In practice we used two types of kernel; the gaussian, and the $\alpha$-decay kernel based on a KNN graph. In the supplementary material, we added two quantitative experiments to compare the use of these kernels and their parameters in Tables 3 and 5. While the performance of the kernel varies, it is always better than the Euclidean distance.
> >
> > An advantage of our distance is that it is multiscale, so it is more robust to a poor parametrization, since by powering the matrix we consider more and more neighbors on the graph defined by the Gaussian or $\alpha$-decay kernel.
> >
> > > How does a value of alpha influence the experimental results?
> >
> > Our goal is to use a ground distance that is equivalent to the geodesic on a Manifold, that is why we need to choose $\alpha\to 1/2$. In practice, we set $\alpha=1/2$. It is true that other choice of $\alpha$ might be interesting, but it is not the scope of this project.
> >
> > > It is unclear to what extent the diffusion geodesic distance approximates anything accurately. Any comments on the accuracy of approximation in Th. 2? Also, the approximation here is stochastic. So the formulation should be either w.r.t. probability or a.s. Any comments about this?
> >
> > This distance has been well established in previous work by Coifman & Leeb who prove the equivalence between $D_\alpha$ and the geodesic $d^{2\alpha}_\mathcal{M}$ and include it in their optimal transport framework. It has been further established by Tong et al. in Diffusion EMD [2,3] for use on data. In general diffusion distances have a long and classic history starting from the seminal paper of Coifman and Lafon who showed that in the limit of infinite data, this distance looks at the difference in heat flow starting from different points in data.
> >
> > As for the formulation of Thm. 2, the formulation is not stochastic and so it is not w.r.t. a probability or a.s. We consider a fix dataset $\mathsf{X}$, we do not consider the generalization to a new sample. The results is about the asymptotic convergence for these points. We modified the phrasing of the Thm.2 to make this clearer.
> >
> > > The geodesic distance is introduced in 139, but there is no exact definition.
> >
> > We use the geodesic as defined in Riemannian geometry. We added a brief intuition behind the geodesic distance. Intuitively, it is the shortest path between two points on the manifold.
> >
> > >153: mu is an actual distribution on M. Is it connected somehow with mu in (3)?
> >
> > No, they are not related. Thank you for pointing this out, we changed the notation in the definition of the kernel.
> >
> > > How does the choice of beta in 153 influence the experimental results?
> >
> > Our approximation of the distance $D_\alpha$ relies on the approximation of the heat kernel. For $\beta=1$, the kernel $P_{\epsilon,\beta}^{t/\epsilon}$ converges to the heat kernel. This is why, in practice, we only use beta = 1. Comparing the anisotropic ($\beta=1$) and isotropic ($\beta=0$), would be interesting, but it is not the focus of this paper.
> >
> > > Why did the authors use Neural ODE? Why not standard RNN? Any rationale behind the choice?
> >
> > Yes, because we want continuous dynamics, to be able to interpolate or extrapolate at any timepoint, not only at a fixed discretization.
> >
> > > Fig. 1: “With the combination of noise, a neural network learns to predict the derivative x′(t0), and an ODE Solver produces the predicted x(t1).” - x(t) is a stochastic process. Which derivative do the authors mean? Also, nothing was told about using the derivatives w.r.t the trajectories in the paper's main text.
> >
> > The caption of Fig.1 wasn’t clear and we rewrote it. It is the derivative of the process with respect to time, there are no derivative w.r.t. to the trajectories.

---

> > > ### Author Response · Authors · 2022-08-02
> > > **Response to Reviewer CfAn part 3**
> > >
> > > > Fig. 1: “Geodesic Autoencoder learns the latent space of representation of data is learnt such that is preserves a diffusion geodesic distance. ” - is this crucial for the considered applications?
> > >
> > > Yes, it is crucial for our applications. As discussed above, cellular systems exhibit manifold structure. Prior work has found diffusion geometry very useful in this application [6,7,8,9]. Our results also confirm the usefulness of diffusion geodesic distances in interpolating flows.
> > >
> > > We provided results to compare the trajectories with and without the Geodesic Autoencoder in Fig. 8 and Fig. 10 in the supplementary material. These figures clearly show that without the Geodesic autoencoder, interpolated paths deviate from the manifold.
> > >
> > > We added Table 5. which quantifies how close the trajectories are to the manifold.  Here with the GAE the average distance of the furthest trajectories is 15.427 (with $\epsilon=0.5$) vs. without the GAE at 21.589. For the experiments, the trajectories using the GAE are always closer to the manifold no matter the kernel as compared to the one using the Euclidean distance.
> > >
> > >
> > > > 182: “With a slight abuse of notation, a similar distance between distributions (rather than points) is constructed in [37] where it was applied to approximate the Earth Mover’s Distance with a geodesic ground cost.” What is the novelty of this approach?
> > >
> > > We rewrote that sentence, because it was misleading. They are not the same distances; however they are similar since they are derived from theoretical results in Leeb and Coifman (2016) They also both rely on an approximation of the heat kernel. The distance in [3], is also equivalent to the Wasserstein with ground distance $D_\alpha$. Our method can be seen as the dynamic counterpart.
> > >
> > > > 189: “we use it to compute distances between predicted and ground truth observations.” Which ground truth observations?
> > >
> > > Say we hold-out sample $\mathsf{X}_2$ and we learn the trajectories from $\mathsf{X}_1$ to $\mathsf{X}_3$. Then $\mathsf{X}_2$ are the ground truth observations. Using the learned trajectories, we interpolate between the timepoints $t=1$ and $t=3$ to predict the sample at $t=2$. We note the interpolated points as $\hat{\mathsf{X}}_2$. To evalute our method, we compute the distance between the interpolated point $\hat{\mathsf{X}}_2$ and the held-out $\mathsf{X}_2$ (our ground truth). In our method, both the geodesic autoencoder and the Neural ODE would be trained without $\mathsf{X}_2$.
> > >
> > > > 191: “Empirically, we observed that G fails to capture meaningful distances for a large number of observations. ” - if it fails, then what is the point of using it?
> > >
> > > Note that $G$ is our realization of $D_\alpha$, and in particular $G$ hides a fixed maximum scale parameter $K$ (as discussed in the supplement). If we take large enough $K$ then $G$ will capture meaningful distances. However, with a fixed $K$, as $n \to \infty$ our approximation of $D_\alpha$ worsens. The problem occurs when $G$ is computed on the **entire** dataset. By encoding the distances $G$ into an autoencoder we can alleviate the computational problem since we use batches of the data to compute $G$ and train the autoencoder.
> > >
> > >
> > > This sentence is overly confusing and not central to our argument, and so we have removed it in the updated draft and replaced it with "Computing $G_\alpha$ on the entire dataset is inefficient due to the powering of the diffusion matrix. We circumvent this difficulty with the encoder, since we train on subsamples."
> > >
> > >
> > > > 202: “to understand how specific genes evolved over time, by decoding in the gene space.” - I do not understand the paper's main topic then. Do the authors propose a new method? Or is this about the modeling of genes? or what?
> > >
> > > Yes, we propose a new method. In short, we define the diffusion geodesic distance $G$, and we learn an embedding that preserves $G$. We then do dynamic optimal transport in that embedding, which is equivalent to the Wasserstein with diffusion ground distance $D_\alpha$ (as well as $d^{2\alpha}_{\mathcal{M}}$). In theory the dynamic optimal transport formulation is valid only for the Euclidean distance, the embedding we learn allows us to extend dynamic OT to other type of ground distances. This can be used for any high dimensional data, presumably sampled from a lower dimensional manifold. We give an example with the cellular trajectories.
> > >
> > > > Th. 3. Which D should we consider? Is any D OK?
> > >
> > > Any dissimilarity respecting the identity of indiscernibles, i.e. $D(\mu, \nu) = 0$ if and only if $\mu=\nu$. We added this requirement in the proof.
> > >
> > > > (7): How do the authors differentiate L_m w.r.t. parameters of the Neural ODE?
> > >
> > > We use the same method as in [4] with the Python Optimal Transport package. We compute the optimal transport plan matrix, and differentiate through the total cost `torch.sum(transport * ground_cost)`. To differentiate w.r.t. the parameters of the ODE we use the adjoint method from [5].

---

> > > > ### Author Response · Authors · 2022-08-02
> > > > **Response to Reviewer CfAn part 4**
> > > >
> > > >
> > > > >260: “trajectories on the manifold”. Why are they located on any manifold? How should the reader analyze Fig. 2 and conclude that the results for the proposed method are better? What is the ground truth? Any accuracy characteristics to compare methods?
> > > >
> > > > We modified the figures 2 and 3 so that it is easier to identify the ground truth, the predicted values and the trajectories. Our method is better in the sense that it stays closer to the manifold. In Tab 4. We quantitatively confirm this observation by looking at the 1-NN distance to the ground truth.
> > > >
> > > > >Table 1: Are the numbers in the table big or small? We should compare this with some simple baseline approaches. Possibly all methods provide some bad results?
> > > >
> > > > We added a baseline distance to give an idea of the magnitude of the numbers. For sample $\mathsf{X}_t$ the baseline distance is the mean between $W_1(\mathsf{X}_{t-1}, \mathsf{X}_t)$ and $W_1(\mathsf{X}_{t+1}, \mathsf{X}_t)$, i.e. the mean of the EMD between the hold-out sample and the previous and next samples.
> > > >
> > > > We currently have the baseline for the Dyngen dataset, we will add it to our other hold-out experiments.
> > > >
> > > > > Appendix, 519: “Replacing the continuity equation by the path dynamic, we have…” Why can we make such a replacement? Any justification that their proof is valid for any dissimilarity?
> > > >
> > > > We revised that sentence since it was a bit confusing. The integral w.r.t. The density $\rho_t$ can be seen as an expectation, where the random variable is $X_t\sim\rho_t$. It is an equivalent formulation, it is also used in [10,11,12]. Any dissimilarity respecting the identity of indiscernibles.
> > > >
> > > > > Appendix, 520-521: Here, the authors wrote that the autoencoder approximates the geodesic distance for sample points. First, this is not completely true, as in Th. 2 we also have a parameter alpha. Also, if these quantities are equal for sample points, it does not follow that for any pair of points from some distribution, the autoencoder will have good generalization ability so that inequalities in 522 are valid.
> > > >
> > > > Here, we only consider sample points, we changed the notation to make things clearer, i.e. we precise $x_i,x_j\in\mathsf{X}$. So there is no generalization involve. As you pointed out, we forgot the $\alpha$ parameters. We changed the notation accordingly, we now use $G_\alpha$ and added the $\alpha$ in the proof.

---

> > > > > ### Author Response · Authors · 2022-08-02
> > > > > **Response to Reviewer CfAn References**
> > > > >
> > > > > [1]. Justin Solomon, Fernando De Goes, Gabriel Peyré, Marco Cuturi, Adrian Butscher, Andy Nguyen, Tao Du, and Leonidas Guibas. Convolutional wasserstein distances: Efficient optimal transportation on geometric domains. ACM Transactions on Graphics, 2015.
> > > > >
> > > > > [2] Alexander Tong, Guillaume Huguet, Dennis Shung, Amine Natik, Manik Kuchroo, Guillaume Lajoie, Guy Wolf, and Smita Krishnaswamy. Embedding signals on graphs with unbalanced diffusion earth mover’s distance. In ICASSP 2022-2022 IEEE ICASSP, pages 5647–5651. 2022.
> > > > >
> > > > > [3] Alexander Y Tong, Guillaume Huguet, Amine Natik, Kincaid MacDonald, Manik Kuchroo, Ronald Coifman, Guy Wolf, and Smita Krishnaswamy. Diffusion earth mover’s distance and distribution embeddings.ICML, pages 10336–10346. PMLR, 2021.
> > > > >
> > > > > [4] Kilian Fatras, Thibault Séjourné, Nicolas Courty, Remi Flamary. Unbalanced minibatch Optimal Transport; applications to Domain Adaptation. ICML 2021.
> > > > >
> > > > > [5] Ricky T. Q. Chen, Yulia Rubanova, Jesse Bettencourt, David Duvenaud. Neural Ordinary Differential Equations. NeurIPS. 2018.
> > > > >
> > > > > [6] Daniel B Burkhardt, Jay S Stanley, Alexander Tong, Ana Luisa Perdigoto, Scott A Gigante, Kevan C Herold, Guy Wolf, Antonio J Giraldez, David van Dijk, Smita Krishnaswamy.  Quantifying the effect of experimental perturbations at single-cell resolution. Nature Biotechnology. 2021.
> > > > >
> > > > > [7] David van Dijk, Roshan Sharma, Juozas Nainys, Kristina Yim, Pooja Kathail, Ambrose J. Carr, Cassandra Burdziak, Kevin R. Moon, Christine L. Chaffer, Diwakar Pattabiraman, Brian Bierie, Linas Mazutis, Guy Wolf, Smita Krishnaswamy, Dana Pe’er. Recovering Gene Interactions from Single-Cell Data Using Data Diffusion. Cell. 2018
> > > > >
> > > > > [8] Kevin R. Moon, David van Dijk, Zheng Wang, Daniel Burkhardt, William Chen, Antonia van den Elzen, Matthew J Hirn, Ronald R Coifman, Natalia B Ivanova, Guy Wolf, Smita Krishnaswamy. Visualizing structure and transitions in high-dimensional biological data. Nature Biotechnology. 2019.
> > > > >
> > > > > [9] Kevin R. Moon, Jay S. Stanley, Daniel Burkhardt, David van Dijk, Guy Wolf, Smita Krishnaswamy. Manifold learning-based methods for analyzing single-cell RNA-sequencing data. Current Opinion in Systems Biology. 2018.
> > > > >
> > > > > [10] Toshio Mikami. Monge’s problem with a quadratic cost by the zero-noise limit of      h-path processes. Probability theory and related fields. 2004
> > > > >
> > > > > [11] Toshio Mikami, Michèle Thieullen. Duality theorem for the stochastic optimal control problem. Stochastic processes and their applications. 2006.
> > > > >
> > > > > [12] Nassif Ghoussoub, Young-Heon Kim, Aaron Zeff Palmer. A solution to the Monge transport problem for Brownian martingales. Annals of Probability. 2021.
> > > > >
> > > > > [13] Gal Mishne, Uri Shaham, Alexander Cloninger, and Israel Cohen. Diffusion nets. Applied and Computational Harmonic Analysis, 2019
> > > > >
> > > > > [14] Andrés F Duque, Sacha Morin, Guy Wolf, and Kevin Moon. Extendable and invertible manifold learning with geometry regularized autoencoders. IEEE International Conference on Big Data. 2020.

---

> > > ### Comment · Reviewer_CfAn · 2022-08-07
> > > **"As for the formulation of Thm. 2..."**
> > >
> > > In the proof of Th. 2 you consider approximation of P by $\mathbf{P}$ using the law of large numbers w.r.t. $n\to\infty$. The formulation of Th. 2 is for a finite $N := min(K, n)$. So could you please comment about the formulation of Th. 2 taking into account this source of stochasticity?

---

> > > > ### Comment · Reviewer_CfAn · 2022-08-09
> > > > **"As for the formulation of Thm. 2..."**
> > > >
> > > > Dear colleagues, any comments?

---

> > > > > ### Author Response · Authors · 2022-08-09
> > > > > **"As for the formulation of Thm. 2..."**
> > > > >
> > > > > >In the proof of Th. 2 you consider approximation of P by $\mathbf{P}$ using the law of large numbers w.r.t. $n\to\infty$. The formulation of Th. 2 is for a finite $N:=min(K,n)$. So could you please comment about the formulation of Th. 2 taking into account this source of stochasticity?
> > > > >
> > > > > Thank you for pointing this out, and making this theorem more precise. We modified the statement of Thm.2 to account for the source of stochasticity:
> > > > >
> > > > > >Assuming $\mathsf{X}$ is sampled from a closed Riemannian manifold $\mathcal{M}$ with geodesic $d_\mathcal{M}$. Then, for all $\alpha\in(0,1/2)$, sufficiently large $K,N$ and small $\epsilon>0$, we have *with high probability* $G_\alpha(x_i,x_j) \simeq d_\mathcal{M}^{2\alpha}(x_i,x_j)$ for all $x_i,x_j\in\mathsf{X}$.
> > > > >
> > > > > We did a similar modification for Cor.1. For the notion of *high probability*, we added two references [15,16] that define high probability bounds on the approximation accuracy of the heat operator, precise bounds are beyond the scope of this work.
> > > > >
> > > > > Please let us know if you have any other suggestions. We will soon upload the corrected manuscript.
> > > > >
> > > > >
> > > > > [15] Xiuyuan Cheng and Nan Wu. Eigen-convergence of gaussian kernelized graph laplacian by manifold heat interpolation. Applied and Computational Harmonic Analysis, 2022
> > > > >
> > > > > [16] A. Singer. From graph to manifold Laplacian: The convergence rate. Applied and Computational Harmonic Analysis, 2006.

---

> > > > > > ### Comment · Reviewer_CfAn · 2022-08-09
> > > > > > **problems with formal statements**
> > > > > >
> > > > > > Dear colleagues,
> > > > > >
> > > > > > 1) It is interesting that earlier in your answer, you wrote, "As for the formulation of Thm. 2, the formulation is not stochastic and so it is not w.r.t. a probability or a.s."
> > > > > >
> > > > > > This looks strange, as it is evident even after first glancing at the proof of the theorem that you consider some equality valid only with some probability.
> > > > > >
> > > > > > So now, in the formulation, it is written: "with high probability". Is 0.01 high enough probability? If not, what is about 0.3? What if, even in the limit, the probability is only 0.1? Then, in this case, does the statement support anything?
> > > > > >
> > > > > > I want to say that current formulations of the theorem and the corollary are not sufficiently strict. For the current paper, this means you should consider the corresponding claims and their proofs not as strict theorems but as plausible mathematical reasonings with appropriate notifications in the main text.
> > > > > >
> > > > > > 2) The authors comment on my other concerns. I am OK with some comments; I am not entirely OK with others. E.g., I still consider the motivation as limited to some extent. The primary motivation is based mainly on the argument that it is important to use a manifold assumption for the cell data, and the Geodesic autoencoder is necessary for this.
> > > > > >
> > > > > > Actually, a) I would propose to mention at least one more applied example, b) demonstrate that without geodesic requirement (i.e., we use some standard AE), we get worse results (kind of an ablation study).
> > > > > >
> > > > > > Conclusion: I will somewhat increase the grade since the authors addressed some my concerns.
> > > > > > But unfortunately, the paper still needs some polishing, corrections, and improvements.

---

> > > > > > > ### Author Response · Authors · 2022-08-09
> > > > > > > **Ablation Studies**
> > > > > > >
> > > > > > > Thank you for your feedback.
> > > > > > >
> > > > > > > > demonstrate that without geodesic requirement (i.e., we use some standard AE), we get worse results (kind of an ablation study).
> > > > > > >
> > > > > > > We note that we have included an ablation study showing that without the geodesic requirements we get worse results. This is shown qualitatively in Figures 8 and 10 and quantified in Tables 3 and 5. For a variety of kernels the geodesic constraint improves performance. We hope these clarify the motivation.

---

### Author Response · Authors · 2022-08-02
**Summary to all reviewers**


We thank the reviewers for their time. We summarize the changes here before responding to each reviewer's individual concerns.

**Summary of Changes**

* **Improvements**
    * Motivation on manifold learning and how the problem of interpolating flows on manifolds arises, particularly in single cell data
    * Enhanced quantitative results comparing MIOFlow with and without the Geodesic Autoencoder and the choice of kernels (Tab.3 and Tab.5).
    * Improved figures (Fig. 2 and Fig. 3) with thicker lines and better legends.
    * Updated Fig.1 on the MIOFlow pipeline and the caption.
    * Smoother transition between Sections 3 and 4.
    * Additional section on Manifold Learning background.

* **EMD instead of MMD quantification** The $W_1$ (EMD) evaluation is correct and is now used throughout. However, during the rebuttal period we found a bug in our maximum mean discrepancy evaluation code, thus we have removed those numbers from the revised paper. Note that these do not alter our conclusions, and were actually inconsistent with the qualitative results in some cases. Those MMD numbers were calculated with Gaussian kernel-MMD optimized as a loss function with a bandwidth determined by the average distance between points, making the numbers incomparable between methods. We will update the paper (and respond here) with correct MMD (with fixed bandwidth) numbers as soon as they are available. However the EMD quantification does not suffer from this and continues to validate our results.  **Edit: We fixed the MMD quantification, see the comment below.**

---

> ### Author Response · Authors · 2022-08-06
> **Quantitative results**
>
> We fixed the problem with the MMD. We report two types of MMD, one with a Gaussian kernel for a fix bandwidth, and one with the identity map ($L^2$ between the means). We re-evaluated both the MMD and $W_1$ (EMD). All of our conclusions are the same, MIOFlow is more accurate than the other methods.
>
> In Tab. 3, we added a comparison between using the GAE or not, for all timepoints and metrics, the accuracy is better with the GAE. We updated all these results in the manuscript.

---

### Meta-Review · Area_Chair_WkLt · 2022-08-26

**Recommendation:** Accept
**Confidence:** Certain

**Metareview:**

Even though several concerns have been raised by multiple reviewers, the reviewers largely agreed on the significance and novelty of the contributions. The authors provided quite detailed responses and given all the data, overall I believe the strengths of the paper outweigh its weaknesses. Hence I am recommending an acceptance.

**Award:**

No

---

### Decision · Program_Chairs · 2022-09-14

Accept